# A circuit suppressing retinal drive to the optokinetic system during fast image motion

Adam Mani ®[1], Xinzhu Yang ®[1], Tiffany A. Zhao[1], Megan L. Leyrer ®[1], Daniel Schreck[1] & David M. Berson[1] ✉

Optokinetic nystagmus (OKN) assists stabilization of the retinal image during head rotation. OKN is driven by ON direction selective retinal ganglion cells (ON DSGCs), which encode both the direction and speed of global retinal slip. The synaptic circuits responsible for the direction selectivity of ON DSGCs are well understood, but those sculpting their slow-speed preference remain enigmatic. Here, we probe this mechanism in mouse retina through patch clamp recordings, functional imaging, genetic manipulation, and electron microscopic reconstructions. We confirm earlier evidence that feedforward glycinergic inhibition is the main suppressor of ON DSGC responses to fast motion, and reveal the source for this inhibition−the VGluT3 amacrine cell, a dual neurotransmitter, excitatory/inhibitory interneuron. Together, our results identify a role for VGluT3 cells in limiting the speed range of OKN. More broadly, they suggest VGluT3 cells shape the response of many retinal cell types to fast motion, suppressing it in some while enhancing it in others.

Image-stabilizing reflexes supporting vision are extremely widespread among animals. In vertebrates, rotation of the head triggers vestibular and visual feedback signals to brainstem and cerebellum oculomotor networks. These generate eye counter-rotation to cancel the global drift or 'retinal slip' of the image. These innate responses comprise the vestibulo-ocular reflex (VOR) and the optokinetic reflex (OKR)[1–3]. Both produce nystagmus, featuring slow image-stabilizing eye movements periodically interrupted by fast resets of gaze in the opposite direction.

Vestibular and visual image-stabilizing reflexes complement one another in the velocity domain. The VOR nulls most of the retinal slip during rapid rotatory head motion; the modest and slow residual slip triggers the OKR, improving the stabilization. During slow head rotation, when VOR gain is weak, OKR is the dominant stabilization mechanism. These two complementary reflexes sum to almost perfectly stabilize the retinal image over the physiological range of head rotations. The OKR is speed-tuned. It must be responsive to slow slip velocities to compensate for the limitations of the VOR, but must be insensitive to fast speeds lest it interfere with fast eye movements, such as saccades or the fast resetting phase of nystagmus.

Among the ~40 types of mammalian retinal ganglion cells (RGCs)[4–6], only a very specific subset is dedicated to visually driven reflexive image stabilization. These comprise a specific subclass of direction-selective ganglion cells (DSGCs), the ON DSGCs. They send their axons almost exclusively to the nuclei of the accessory optic system, which relays their retinal slip signals to the vestibulocerebellum and brainstem oculomotor centers[1,2,7]. This circuit is highly conserved across vertebrates, including apparently homologous ON DSGCs[8]. The far more common ON-OFF class of DSGCs makes only a small contribution to this circuit, projecting instead mainly to the lateral geniculate nucleus and superior colliculus.

ON DSGCs, like the OKR they drive, respond best to slow speeds of retinal slip, typically ~1°/s, depending on species and stimulus[2,9,10]. They are effectively silent during fast visual motion, as occurs during saccades or the fast phase of nystagmus. By contrast, ON-OFF DSGCs respond well to such fast motion[10] (but see ref. 11). Other types of RGCs exhibit diverse speed tuning profiles, with some highly responsive to very fast motion.

ON DSGCs thus encode both components of the vector of retinal slip−speed and direction−as required for image stabilization. Starburst amacrine cells (SACs) confer direction selectivity on these and other DSGCs through well-studied synaptic mechanisms[12–15]. By contrast, much less is understood about the cell types, neurotransmitters, and synaptic circuits that confer slow speed tuning upon ON DSGCs or shape the speed preferences of other RGC types.

[1]Department of Neuroscience, Brown University, Providence, RI, USA. ✉e-mail: david_berson@brown.edu

Diverse mechanisms could generate slow-speed tuning in ON DSGCs. Excitatory drive might falter at higher speeds due to sub-optimal spatiotemporal summation of excitatory inputs across the dendritic arbor[16] or intrinsic filtering of the high-frequency excitatory events induced by fast motion. Alternatively, fast motion might trigger inhibition. This could act presynaptically, by suppressing the cell's glutamatergic inputs from bipolar cells and from VGluT3 amacrine cells[17]. Alternatively, fast motion could evoke direct feedforward inhibition onto the ON DSGC itself. Indeed, this is the key mechanism for slow-speed preference in ON DSGCs of rabbits, with glycine serving as the main inhibitory transmitter[10]. ON-OFF DSGCs, a closely related class of direction-selective neurons, lack this glycinergic input and respond well to fast speeds[10].

Here in mouse retina we have used serial block face electron microscopy (SBEM), patch recordings, dendritic calcium imaging, and cell-type-specific optogenetics and chemogenetics to delineate the synaptic networks responsible for vetoing ON DSGC responses at high speeds. Our observations in mice echo those in the rabbit: direct feedforward glycinergic inhibition of ON DSGCs is primarily responsible[10]. Our connectomics studies identify an unexpected source of this glycinergic inhibition. It is the VGluT3 amacrine cell, an unusual interneuron that makes excitatory (glutamatergic) synapses onto some postsynaptic targets and inhibitory (glycinergic) synapses onto others[18–20]. Our functional data confirm that VGluT3 cells are the source of glycinergic inhibition onto ON DSGCs during fast motion. They also show that glutamatergic inputs from VGluT3 cells to ON-OFF DSGCs (the other directional ganglion-cell class) augment their responses to fast motion. The connectomic data document widespread VGluT3 output to most types of RGCs. These amacrine cells may therefore play a major role in sculpting retinal output during fast motion by enhancing the responses of some RGCs and inhibiting others.

## Results

### Inhibition at fast speeds generates slow speed tuning in mouse ON DSGCs

To explore the speed-tuning mechanism in ON DSGCs, we made patch recordings of their synaptic currents and spiking responses while full-field gratings moved in the preferred direction at various speeds. We located these rare RGCs using 2-photon imaging of ON-DSGC-selective fluorescent reporters (Hoxd10-GFP[2] or Pcdh9-Cre[21]) or by surveying RGCs for characteristic extracellular spike responses to flashed stimuli (see Supplementary Note 1 and "Methods"). We confirmed the identity of ON DSGCs by their selectivity for the direction of grating drift and through dye-filling and imaging of their stereotyped dendritic morphology (Supplementary Fig. 1).

As expected, ON DSGCs were remarkable among RGCs in preferring slower speeds of grating drift (Fig. 1a, b, d, blue). Spiking was strongest at retinal speeds of $150 \pm 11\,\mu m/s$ (angular velocity of $5 \pm 0.4°/s$; mean ± SEM; $n = 63$), close to previously reported optima for ON-DSGC responses and OKN gain in mice[2]. This probably slightly overestimates the mean preferred speed: About a quarter of the recorded cells responded best to the slowest speed tested ($76\,\mu m/s$) and may have preferred even slower speeds. Spike responses were strongly attenuated at speeds above the optimum, dropping to half their maximum spike rate at roughly twice the preferred speed ($360 \pm 23\,\mu m/s$, $12 \pm 0.8°/s$, $n = 63$, see "Methods").

Many other RGC types responded to much higher speeds of grating drift (Fig. 1d). For example, ON-OFF DSGCs exhibited optimum speeds about four times faster than those of ON DSGCs ($600 \pm 70\,\mu m/s$; half max. speeds $2000 \pm 160\,\mu m/s$, $67 \pm 5°/s$ ($n = 17$), Fig. 1d). Some RGC types responded to even higher speeds. For example, ON alpha cells maintained better than half-maximal responses even above $3000\,\mu m/s$ ($100°/s$). Other types, such as ON-delayed RGCs[22], shared the slow speed tuning of ON DSGCs.

We hypothesized that the slow-speed preference of mouse ON-DSGCs results from direct glycinergic inhibitory input, as demonstrated in rabbit retina[10]. To test this idea, we first used whole-cell voltage clamp to record excitatory and inhibitory currents evoked in ON DSGCs by various speeds of grating drift in the preferred direction (Fig. 1a–c, Supplementary Fig. 2a, red, orange curves). Both excitatory and inhibitory inputs were broadly tuned for speed, but with different profiles. Excitation was present at the slowest speeds tested, but rose with speed, peaking at speeds five times faster than those evoking the most robust spiking ($710 \pm 90\,\mu m/s$ ($n = 11$), or $24°/s$; peak charge transfer $490 \pm 70$ pC). Excitation declined to half its maximum at $2290 \pm 170\,\mu m/s$. Thus, excitatory drive is tuned for speed, but with a very different profile than for spiking, implicating other mechanisms in the speed tuning of ON-DSGC output.

Though inhibition was strongest at speeds close to those evoking the strongest excitation ($1220 \pm 170\,\mu m/s$, declining half max. $>2900\,\mu m/s$, $n = 12$; peak charge transfer $2200 \pm 230$ pC), it was weaker than excitation at slow speeds and stronger than excitation at fast speeds. The two normalized curves crossed at ~$700\,\mu m/s$. At speeds faster than this, nearly all spiking was suppressed (Fig. 1a). The ratio between excitation and inhibition (E/I ratio) sank below ~0.25 at the point of crossover (absolute charge transfer; Supplementary Fig. 2b). In ON-OFF DSGCs, inhibition ~3–3.5 times as large as excitation is sufficient to suppress RGC firing[23,24]. Therefore, inhibition appears to account for the response behavior with no need to invoke intrinsic mechanisms, such as an unusually high spiking threshold. These findings imply that slow speed tuning of ON DSGCs reflects a marked reduction in E/I ratio at higher speeds, as in the rabbit[10].

For most speeds, excitatory current exhibited a single prominent peak for each cycle of the grating (Fig. 1c, red), presumably reflecting input from ON bipolar cells, while inhibitory current often exhibited minor peaks roughly anti-phase to the excitation. The OFF pathway may therefore make a larger contribution to inhibition than to excitation. At higher speeds, the inhibitory current exhibited not only the stimulus-locked modulatory component but also an underlying continuous inhibitory conductance, again confirming earlier findings in rabbit ON DSGCs.

### Fast retinal slip triggers glycinergic inhibition of ON DSGCs

We repeated these experiments adding receptor antagonists to the bath to test the hypothesis that the feed-forward inhibition evoked by fast motion was glycine-mediated, as in rabbit (Fig. 1e, f). Strychnine, a glycine receptor antagonist, sharply attenuated the inhibition and dramatically reduced the steady plateau of inhibition evoked by fast grating motion. At speeds evoking maximum inhibitory current, total inhibition was reduced by more than half its control value ($57 \pm 4\%$ decrease in charge transfer; $p = 2 \times 10^{-5}$ (one-tailed paired t-test), $n = 3$. In contrast, blockade of GABA$_A$ receptors alone (SR95531) had only a modest effect (Fig. 1e, f; $17 \pm 7\%$ decrease in charge transfer; $p = 0.08$, $n = 3$). Simultaneous application of both antagonists abolished all inhibitory current in the ON DSGCs. Thus, effectively all inhibition driven by fast motion is mediated by GABA$_A$ and glycine receptors under our experimental conditions.

Consistent with the large glycinergic contribution to overall inhibition, strychnine dramatically extended the range of speeds over which motion induced ON-DSGC spiking. This parallels the findings in rabbit[10] and confirms that glycinergic inhibition plays a key role in limiting responses to rapid motion (Fig. 1g, h; an increase at half max. of $1500 \pm 500\,\mu m/s$, $p = 0.018$, $n = 4$). In the presence of strychnine, ON DSGC firing could track the high temporal frequencies associated with fast grating motion (Fig. 1h). This demonstrates that membrane properties of ON DSGCs do not preclude responses to fast motion. The strychnine-mediated increase in firing at higher speeds was not due to increased excitation (Supplementary Fig. 2c). Taken together, these

 

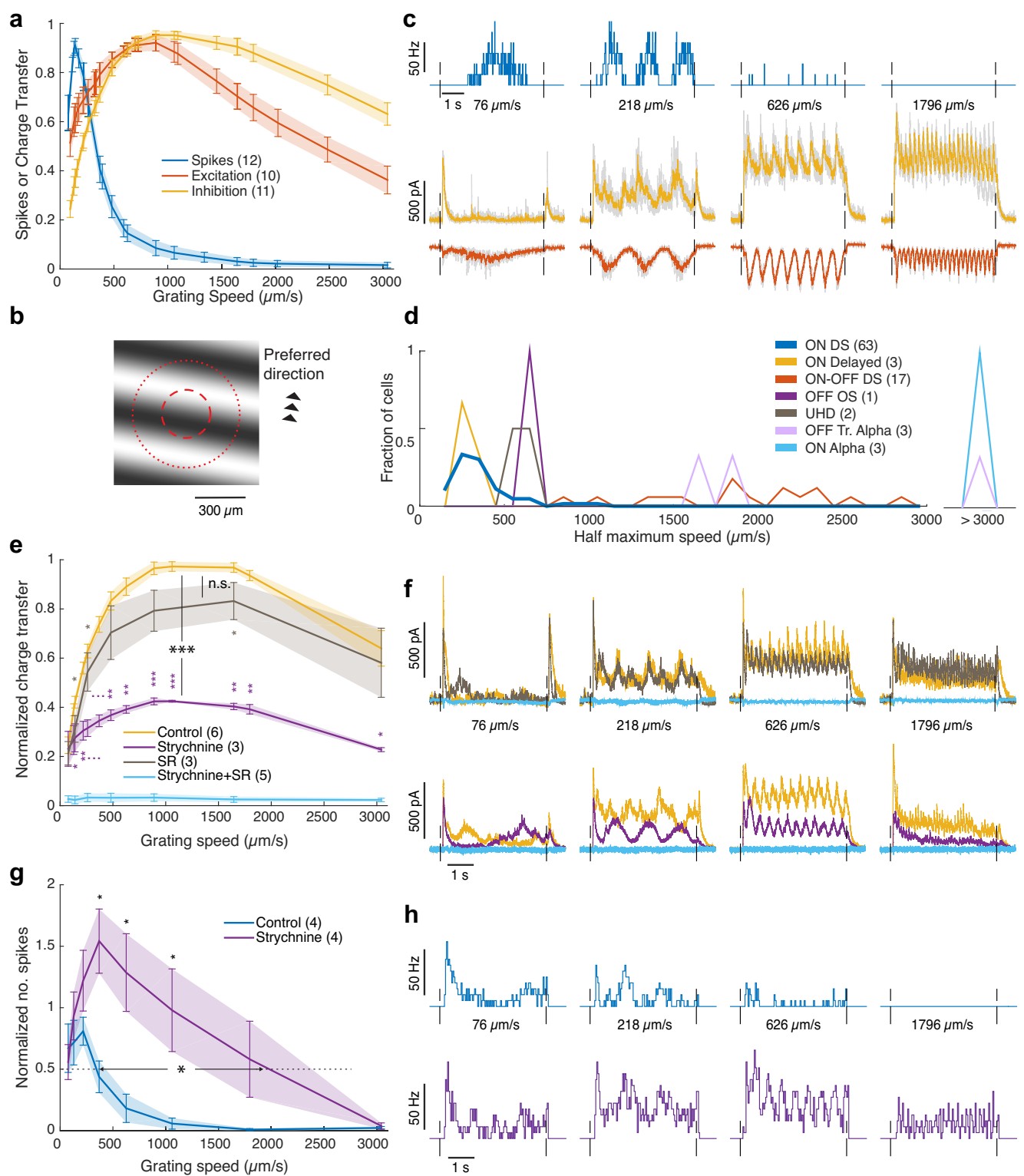

results demonstrate that fast motion triggers direct glycinergic inhibition of ON DSGCs, as first shown in rabbit[10].

## VGluT3 cells are the source of glycinergic synapses onto ON DSGCs

To identify the sources of glycinergic suppression of ON-DSGCs during fast motion, we turned to serial electron microscopic reconstruction. In a published SBEM dataset of the adult mouse inner plexiform layer spanning >200 μm[25] we partially reconstructed the neurons making non-ribbon synapses onto a previously reconstructed ON DSGC[16]

(460 synapses; Fig. 2). Its identity as an ON DSGC is supported by convergent anatomical evidence (see ref. 16 and Supplementary Note 2). The great majority of the amacrine-cell synapses onto this ganglion cell came from starburst amacrine cells (SACs; 83% of 438 synapses). SACs are GABAergic and cholinergic, and not glycinergic. A small minority of the synapses derived from widefield amacrine cells (7%), which are also presumably GABAergic, not glycinergic[26–29]. Virtually all the remaining synaptic contacts came from VGluT3 cells (9%, $n = 40$ synapses from 16 cells), an amacrine-cell type that releases both glutamate and glycine as neurotransmitters.

**Fig. 1 | Inhibitory mechanism underlying slow speed tuning in ON DSGCs.**
**a** Speed tuning in the response of ON DSGCs to moving gratings, as revealed by spiking (blue) and synaptic currents (excitation: red; inhibition: orange). Throughout Fig. 1, curves with error bars and shadings are averages ± SEM over cells; numbers of cells are in parentheses. Cells are normalized by their maxima. Absolute maxima: 96 ± 10 for spikes, 490 ± 70 pC for excitation and 2200 ± 230 pC for inhibition. Currents, expressed as charge transfer, were measured at a holding voltage of −65 mV (excitation) or +20 mV (inhibition). **b** The grating stimulus, with a schematic ON DSGC receptive field (center and surround; red circles) shown to scale. **c** Time course of mean spiking and current responses (3 repeated trials) in an ON DSGC to different grating speeds; color scheme as in (**a**). Gray: individual trials. **d** Distribution of speed preferences among RGCs of different types as indexed by the highest speed capable of evoking a response half that evoked by the optimal speed. **e** Glycinergic blockade (strychnine) strongly reduces inhibition relative to control conditions ($p = 2.2 \times 10^{-5}$, curves' maxima), while blocking ionotropic GABA

receptors (SR95531) had a weaker effect ($p = 0.078$). Combined application blocked all inhibition. Large stars: Significance of difference at maxima between control and drug conditions. Small stars: significance of the difference between curves at specific speeds. (*), (**), (***): $p < 0.05$, 0.01, and 0.001, respectively (none: $p > 0.05$), in a paired one-sided Student's t-test throughout the figure. **f** Effects of GABA blockade (top row) or glycinergic blockade (bottom row) on the inhibitory currents. In both cases, the other antagonist was subsequently added to the first, blocking all inhibition. **g** Effect of glycinergic blockade on ON DSGC firing. The speed range over which the cells responded increased ~4.5 fold ($p = 0.018$). Dotted horizontal line: half maximal control response. The speed difference between individual curves was measured along this line. Large star: significance of the horizontal difference. Small stars: as in (**e**). **h** Effect of glycinergic blockade on the firing. Spiking was able to follow fast gratings. Traces in (**f**), (**h**) were averaged over three repeated trials in a given cell. Source data are provided as a Source data file.

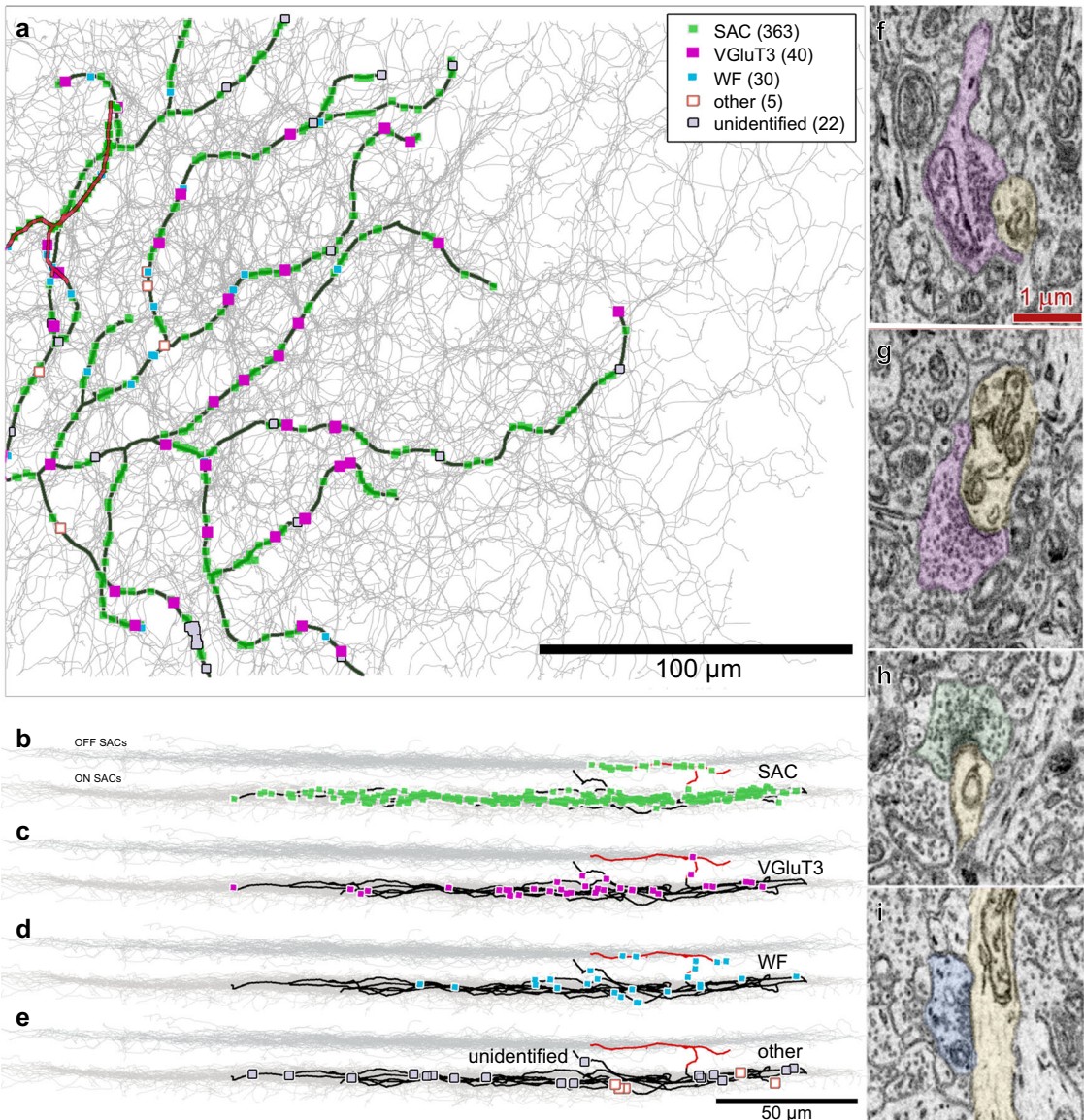

**Fig. 2 | Amacrine cell inputs onto an ON DSGC. a** Electron-microscopic reconstruction of the distribution and sources of all detected amacrine-cell synaptic contacts onto an ON DSGC. Dendrites of the ganglion cell are shown in black except for the sparse branches in the OFF sublayer (red). Synaptic contacts are plotted as dots, color coded by amacrine-cell source (see key), as determined by reconstruction of presynaptic cells. Numbers of synapses are in parentheses. 'Other' synapses: H18 AC (2), medium field unidentified AC (2), and A17 AC (1).

'Unidentified': fragments of dendritic arbors too small to identify. The plexus of ON SAC dendrites is shown in gray. Boundaries of the SBEM volume appear as a fine rectangle. **b**–**e** side (vertical) views of depth of these inputs within the IPL. Gray bands mark the ON and OFF SAC plexuses. Color scheme as in (**a**). **g**–**i** Example SBEM images of amacrine-cell synapses onto the ON DSGC. ON-DSGC dendritic profiles are tinted yellow. The presynaptic cells are color coded as in (**a**–**e**): VGluT3 cells (**f**, **g**), an ON SAC (**h**) and a WF amacrine cell (**i**).

Convergent anatomical evidence supporting the identity of the reconstructed neurons as VGluT3 cells is provided in Supplementary Note 2 and Supplementary Fig. 3, along with additional connectomic observations and technical details. The only other plausible glycinergic contacts onto this ON DSGC were traced to Type H18 amacrine cells[30,31] (2 synapses) or to an unidentified medium-field type (2 synapses). The cellular source of a few remaining contacts (22 of 460 synapses) had too few dendrites within the volume to be identified. We conclude that VGluT3 amacrine cells are by far the best candidate source of glycinergic inhibition shaping the slow-speed tuning of ON DSGCs.

## VGluT3 cells make functional glycinergic synapses onto ON DSGCs

The connectomic analysis revealed that a single type of glycinergic amacrine cell—the VGluT3 cell—supplies virtually all of the glycinergic synaptic contacts onto ON DSGCs. The VGluT3 amacrine cell is thus the presumed source of the glycinergic suppression of ON DSGCs during rapid motion. This is a surprising conclusion because VGluT3 cells have been reported to provide glutamatergic excitation to ON DSGCs, not glycinergic inhibition[17]. To test for the inferred glycinergic influence, we recorded postsynaptic currents evoked in ON DSGCs by selective optogenetic depolarization of VGluT3 cells. For these studies, we crossed a strain of VGluT3-Cre mice with a strain (Ai32) that expresses channelrhodopsin2 (ChR2) in a Cre-dependent manner (see "Methods", Supplementary Note 3, Supplementary Fig. 4). Conventional photoreceptor-mediated light responses and cholinergic transmission were suppressed with a cocktail of synaptic blockers ('photoreceptor block'; L-AP4, ACET, hexamethonium[17]).

Optogenetic activation of VGluT3 cells evoked robust inhibitory currents in practically all ON DSGCs tested. Peak inhibitory postsynaptic currents averaged $43 \pm 7$ pA and $26 \pm 6$ pA, for single light pulses with durations 1 s and 0.1 s, respectively (Fig. 3a, b, Supplementary Fig. 5a; $n = 13$ cells; three cells were excluded due to atypical or inconsistent currents, see "Methods" and Supplementary Note 5). Inhibitory currents were also strongly activated by optogenetic pulse trains matching the temporal frequencies produced by fast grating motion (see Supplementary Note 4 and Supplementary Fig. 5c).

The inhibition evoked by optogenetic activation of VGluT3 cells was glycinergic, as expected. Blocking ionotropic glutamate receptors (CNQX, D-AP5) had little effect on the evoked inhibitory conductance (Fig. 3a, c; mean reduction in maximal current of $21 \pm 5\%$, 4 cells), whereas further blockade of glycine receptors (strychnine) reduced the current to below the noise level (decrease of $78 \pm 6\%$, 4 cells). The broad-spectrum glutamate blockade used here ensures that these evoked currents are mediated optogenetically, not because rods and cones are activated by the photostimulation with subsequent glutamatergic signaling through synaptic networks. It also excludes indirect effects mediated by evoked release of glutamate from VGluT3 cells. We conclude that VGluT3 amacrine cells inhibit ON DSGCs through direct glycinergic synaptic input.

We reversed the order of drug application in three cells. When strychnine was the first inhibitory blocker applied, the optogenetically evoked inhibition was completely suppressed in one cell (Fig. 3d, left, 4e). However, in two other cells, strychnine left at least half the inhibition intact. This residual inhibition was abolished upon further addition of the glutamate receptor antagonists (Fig. 3d, right, 4e). Presumably this strychnine-resistant inhibitory current occurs when evoked glutamate release from VGluT3 activation glutamatergically excites one or more GABAergic amacrine cell types that synapse onto ON DSGCs (see "Discussion").

Control experiments confirmed that the observed currents were due to ChR2 (Supplementary Fig. 6c); that off-target ChR2 expression in Müller cells was not responsible for the evoked currents in ON DSGCs (Supplementary Fig. 6b); that other known targets of VGluT3 input responded as expected to the optogenetic stimulus (Fig. 3f); and

that the pharmacological effects shown in Fig. 3a, c–e were not simply the result of rundown of optogenetic responses over trials (Supplementary Note 5, Supplementary Fig. 6a).

To probe the previously reported glutamatergic excitation of ON DSGCs by VGluT3 cells[17], we repeated these experiments while clamping ON DSGCs at the chloride reversal potential. We detected optogenetically evoked EPSCs, but they only slightly exceeded the noise and typically required longer (1 s) light pulses (Peak currents: $9 \pm 1$ pA, $n = 8$ cells; Fig. 3g, h, Supplementary Fig. 5b, d, e). This excitation was mediated by glutamate: it persisted in the presence of strychnine and was eliminated by blockade of ionotropic glutamate receptors (Fig. 3g, i; $n = 1$). All ON DSGCs exhibiting such evoked excitation also exhibited optogenetically evoked IPSCs when clamped at +20 mV ($n = 8$). Optogenetically evoked IPSC and EPSC measured in the same cells correlated positively (Supplementary Fig. 5d, e; $R^2 = 0.53$, $n = 8$ cells). This is consistent with both currents arising from VGluT3 cells and variable success at driving that dual input optogenetically.

## VGluT3 suppression reduces fast-motion inhibition of ON DSGCs

To test the idea that VGluT3 cells are the source of the inhibition that suppresses ON DSGC during fast motion, we studied the effects of chemogenetic suppression of VGluT3 cells on this inhibition. We used a DREADD system (Designer Receptor Exclusively Activated By Designer Drugs) in VGluT3-Cre mice, expressing the hM4Di receptor in VGluT3 cells either by intraocular injection of a Cre-dependent viral vector or by crossing VGluT3-Cre mice with Cre-dependent DREADD mice ("Methods"). In either case, bath-application of the DREADD ligand (CNO) markedly reduced the inhibitory currents induced by drifting gratings at all speeds (Fig. 4a–e, Supplementary Fig. 7a, b). At speeds evoking maximal inhibition, CNO reduced inhibitory charge transfer by $43 \pm 7\%$ ($p = 0.012$, $n = 3$) for viral experiments and $38 \pm 6\%$ ($p = 0.012$, $n = 3$) for those based on the genetic cross. The same ligand had no effect on inhibition of ON DSGCs in a DREADD-free mouse (Fig. 4c, $p = 0.39$, $n = 3$). Thus, the native inhibition in ON DSGCs at high speeds is provided at least to a large extent by glycine release from VGluT3 cells.

The chemogenetic approach was less effective than strychnine in reducing the motion-induced inhibition (Fig. 4d, e). Additional analysis suggests that this is mainly because our suppression of VGluT3 cells was incomplete, not because other amacrine cell types make major contributions to the glycinergic influence (see Supplementary Note 6, Supplementary Fig. 8a, b and "Discussion"). Chemogenetic suppression of VGluT3 cells did not significantly affect the speed dependence of ON DSGC excitation (Supplementary Fig. 8c, d).

## VGluT3 cells augment fast-motion responses in ON-OFF DSGCs

We found that the other class of direction-selective ganglion cells, the ON-OFF DSGCs, were excited by optogenetic activation of VGluT3 cells (Fig. 3f), as previously reported[17,32]. We saw no sign of inhibition. Thus, VGluT3 cells apparently exert mainly opposing influences on the two DSGC classes. ON-OFF DSGCs responded well to gratings moving at speeds that induced strong glycinergic suppression in ON DSGCs (Fig. 1d). This suggests that glutamate release from VGluT3 cells may augment ON-OFF DSGC responses in this range of speeds. Indeed, chemogenetic suppression of VGluT3 cells reduced the excitation evoked in ON-OFF DSGCs by fast gratings moving in the preferred direction ($17 \pm 3\%$ reduction in excitatory charge transfer from its control value; $p = 0.017$, $n = 3$; Fig. 4f, Supplementary Fig. 7c, d). Spiking responses in ON-OFF DSGCs were similarly reduced by the chemogenetic manipulation (Fig. 4g, Supplementary Fig. 7e, f). Though incomplete blockade of VGluT3 cells may underrepresent these influences, most of the residual excitatory current presumably represents bipolar input. Thus, during rapid visual motion, VGluT3 cells affect the two DSGC classes in opposing ways, enhancing responses in ON-OFF DSGCs while suppressing ON DSGCs.

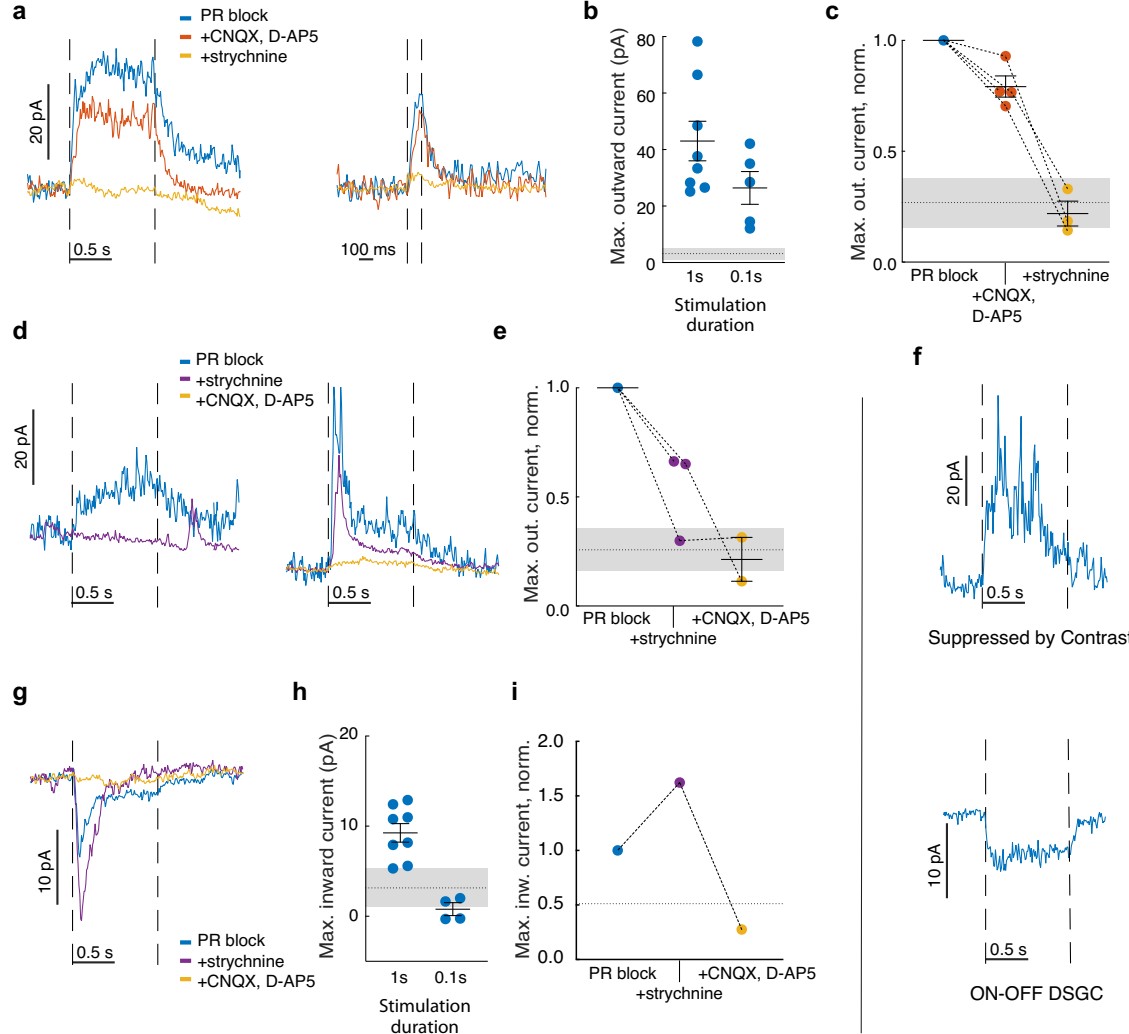

**Fig. 3 | Postsynaptic currents evoked in ON DSGCs by optogenetic activation of VGluT3 cells. a** Inhibitory currents (holding voltage V_hold: +20 mV) in two ON DSGCs (left and right panels) in response to optogenetic stimulation during the period marked by interrupted vertical lines. Blue traces were acquired under photoreceptor (PR) block consisting of L-AP4, ACET and hexamethonium. Pharmacological blockade first of glutamate receptors (red) and then of glycine receptors (orange) show that the current is glycine mediated. Current traces were averaged over 3–5 repeated trials and smoothed. **b** Summary of optogenetically evoked inhibitory (outward) currents in ON DSGCs. Peak currents are shown for stimulation durations of 1 s (8 cells) or 0.1 s (5 cells), along with their mean ± SEM. Throughout Fig. 3, the dotted line and shaded region represent the noise level (mean ± SD over cells, see "Methods"). **c** Summary of 3-step pharmacology experiments where the photoreceptor block was followed by glutamate (CNQX, D-AP5), and then glycine blockade (strychnine). Colors are as in (**a**) (4 cells with

mean ± SEM, peak currents were normalized by the control value). **d** As for panel (**a**), but with glycinergic blockade (purple) preceding glutamatergic blockade. **e** As for panel (**c**), but summarizing experiments with glycine blockade preceding glutamate blockade, as in (**d**) (3 cells). **f** Optogenetic activation of VGluT3 cells evokes inhibitory currents in a suppressed-by-contrast RGC (top panel; V_hold: +20 mV), and excitatory currents in an ON-OFF DSGC (bottom panel; V_hold: −65 mV; both under photoreceptor block). **g** Excitatory currents evoked in an ON DSGC by optogenetic activation of VGluT3 cells and the pharmacological effects on this current in the same cell. V_hold: −65 mV. Colors are as in (**d**). **h** Summary of optogenetically evoked excitatory (inward) currents in ON DSGCs (peak currents, mean ± SEM; 8, 4 cells for 1 s, 0.1 s stimulation, respectively). **i** Peak inward currents in the ON DSGC shown in (**g**). The dotted line represents noise for the cell. Colors in (**e**), (**g**), (**i**) are as in (**d**). Source data are provided as a Source data file.

## VGluT3 dendrites are activated by fast global motion

To provide more direct evidence that fast motion triggers VGluT3 output, we recorded their calcium responses. We selectively expressed GCaMP6 in these cells using a genetic cross (VGluT3-Cre x Ai148)[33].

VGluT3 dendrites exhibited robust calcium signals in response to the motion of full-field grating stimuli, especially at the fast speeds that trigger feedforward glycinergic inhibition of ON DSGCs (Fig. 5a–e, 512 ROIs, 14 FOVs, 4 mice). Speed tuning was highly consistent for different ROIs within each field of view, across most fields of view (FOV; Fig. 5c–e), and at different depths within the VGluT3 plexus. Some FOVs exhibited an unusually strong response to the fastest speeds (Fig. 5d).

The robust response to extended stimuli was unexpected because VGluT3 receptive fields have strong suppressive surrounds[17,33,34]. Indeed, calcium responses were greatly enhanced when we restricted the same gratings to a circular patch centered on the FOV (345 μm diam.; Fig. 5f, Supplementary Fig. 9). This enhancement was apparent at all speeds, and increased the response to the optimal speed ~5-fold ($p = 4 \times 10^{-4}$, one-tailed paired t-test, $n = 4$ FOVs. Each FOV is the average of its ROIs). Slow gratings that failed to drive responses when presented full-field now evoked clear Ca²⁺ signals. Nonetheless, the high-speed preference remained.

To probe the spatial organization of these surround effects, we presented spots or masked gratings of various sizes centered on the

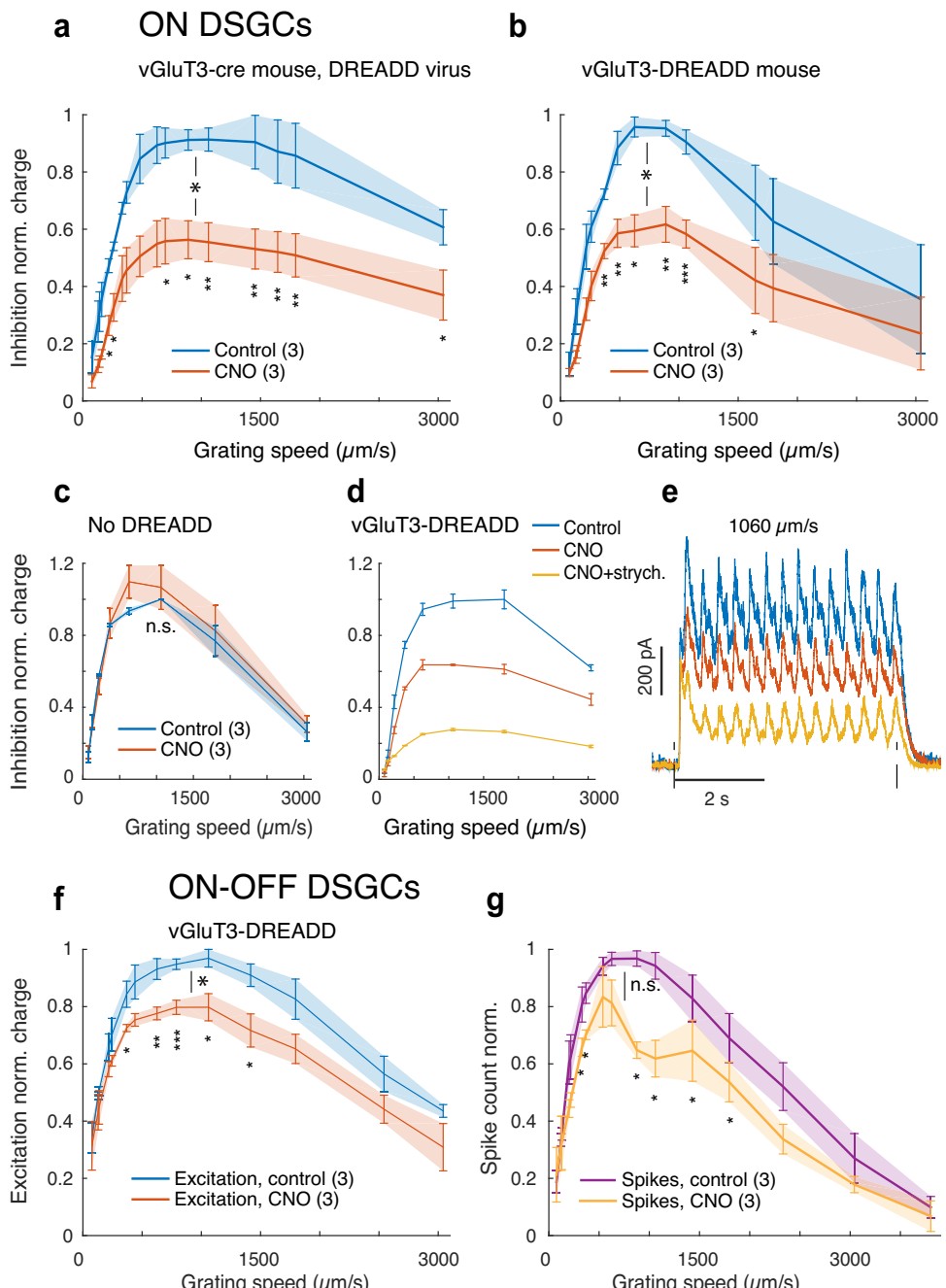

**Fig. 4 | Chemogenetic suppression of VGluT3 cells alters speed tuning of ON and ON-OFF DSGCs. a, b** Application of the DREADD ligand CNO (red) partially suppressed inhibitory currents evoked by moving gratings in ON DSGCs under control conditions (blue), regardless of whether the method for Cre-dependent DREADD expression was intraocular injection of a Cre-dependent virus (**a**; $p = 0.012$) or through a genetic cross (**b**; $p = 0.012$). Voltage clamp, $V_{hold}$: +20 mV. Maximal charge transfers (control): $2350 \pm 460$ pC and $1830 \pm 750$ pC for the virus and cross, respectively. Throughout Fig. 4: unless otherwise stated, curves with error bars and shadings are averages ± SEM over cells; numbers of cells are in parentheses. Large stars or 'n.s.': Significance of difference at maxima between control and drug conditions. Small stars: significance of the difference between curves at specific speeds. (*), (**), (***): $p < 0.05$, 0.01, and 0.001, respectively (none: $p > 0.05$), in a paired one-sided Student's t-test. **c** Control experiment showing that

CNO has no effect when applied in DREADD-free retinas from HoxD10-GFP or C57BL/6 mice (two-sided paired t-test, $p = 0.39$). **d** Most inhibition that remains after CNO application is blocked by further addition of the glycine-receptor antagonist strychnine. The curves represent mean ± SEM over 3 repeated trials in each condition in a single ON DSGC. **e** Current traces of the ON DSGC in (**d**) under the same pharmacological conditions and using the same color scheme as in (**d**). Mean over 3 repeated trials. Grating speed: $1060\,\mu$m/s. **f** Speed tuning profiles of excitatory charge in ON-OFF DSGCs, before and after application of CNO. $V_{hold}$: −65 mV. Max. charge transfer (control): $690 \pm 160$ pC. $p = 0.017$. **g** Speed tuning profiles of action-potential firing in ON DSGCs, before and after application of CNO. Max. spike count: $120 \pm 6$ (5 s). $p = 0.15$, paired one-sided t-test. For experiments in (**f**) and (**g**), DREADD was expressed through a genetic cross. Source data are provided as a Source data file.

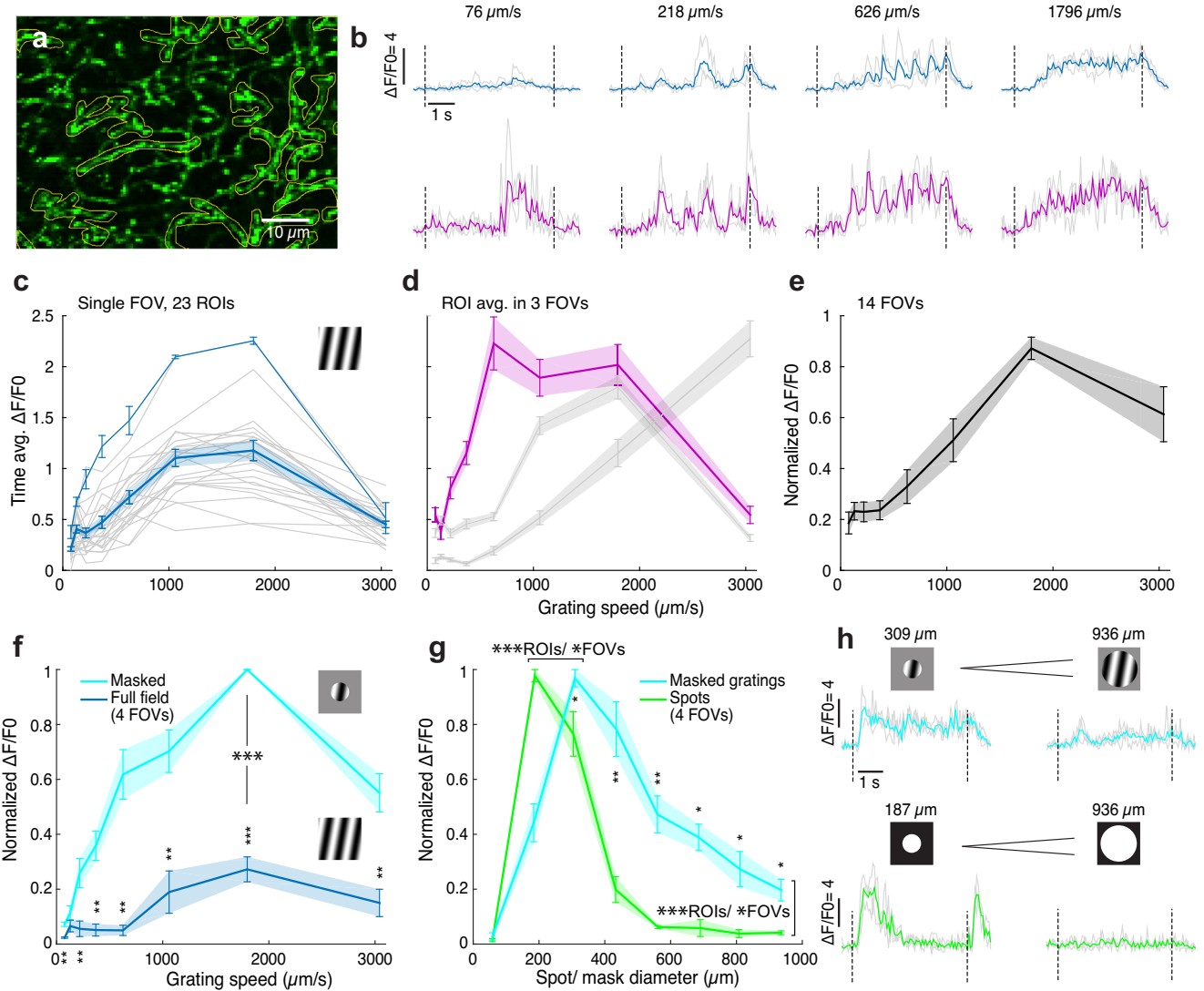

**Fig. 5 | Calcium responses in VGluT3 dendrites. a** GCaMP6 in VGluT3 dendrites. Yellow outlines mark regions of interest (ROIs). **b**–**e** Responses to full-field gratings drifting at different speeds in an arbitrary fixed direction. **b** Fluorescence traces in two ROIs (blue and magenta), grating speeds denoted above. Individual trials, gray; mean over 3 repeated trials, blue or magenta. **c** Calcium response (ΔF/F, time averaged) in 23 ROIs in a given FOV (gray), with their mean ± SEM (thick blue curve). Thin blue curve: single ROI shown in (**b**) in blue (response mean ± SEM over 3 trials). **d** Response mean ± SEM over ROIs in three more FOVs (ROIs: $n = 25, 31, 30$). Magenta: FOV that includes the ROI shown in (**b**) in magenta. **e** Mean ± SEM of normalized response curves from 14 FOVs. Absolute maxima: 1.2 ± 0.17. **f** As for (**c**–**e**), but limiting the extent of the grating by a circular mask (345 μm diameter, light blue). Full-field grating responses were recorded for comparison (blue). (mean ± SEM over 4 FOVs, mean curve from each FOV; absolute max. masked gratings: 3.4 ± 0.6). Large stars: significance for difference at maxima

($p = 3.9 \times 10^{-4}$). In (**f**) and (**g**), small stars denote significance at specific data points: (*), (**), (***): $p < 0.05, 0.01,$ and $0.001$, respectively (none: $p > 0.05$), in a paired one-sided t-test. **g** Area-response functions for spots (green) and gratings (light blue) of different sizes. Gratings were presented within a circular mask of variable size (speed: 1520 μm/s). Absolute maxima: 4 ± 0.7 and 2.3 ± 0.5, for gratings (avg. 5 s) and spots (avg. first 2 s), respectively. Large starts: significance was tested for the difference in optimal response diameter, and the difference in response at the largest diameter tested (surround suppression). Paired, one-sided t-test over 4 FOVs: optimal diameters, $p = 0.012$, suppression, $p = 0.031$. Unpaired, one-sided t-tests over ROIs within a FOV: optimal diameters, $p < 4 \times 10^{-4}$, suppression, $p < 5 \times 10^{-8}$, 28–73 ROIs (see "Methods" for details). **h** Traces from single ROIs in (**g**), in response to gratings (light blue) or spots (green). Stimulus schematics and sizes are shown above. Source data are provided as a Source data file.

FOV (Fig. 5g, h, Supplementary Fig. 10a, b). The gratings were the same regardless of mask size, and moved at a high speed (1520 μm/s; "Methods"). Area-response profiles based on spot stimuli closely resembled those reported earlier based on patch recordings from VGluT3 somas[34] (Fig. 5g, green curve), with strong surround suppression and maximal responses for spot diameters of 219 ± 19 μm (5 FOVs). Grating patches (light blue curve) yielded markedly different area-response profiles, with much less response attenuation when grating patches extended into the surround. For the largest stimuli tested (936 μm diam.) grating responses were reduced to a fifth of their optimal response (20 ± 4%, 4 FOVs), while spot responses were

suppressed to less than a tenth of their maximum (8 ± 2%, 4 FOVs. Difference between spots and gratings: $p = 0.03$; For ROI ensembles within the same FOV, $p < 10^{-7}$). Further, the optimum stimulus size for gratings was 50% larger than for spots (330 ± 35 μm; 5 FOVs; $p = 0.01$; ROIs within FOVs: $p < 4 \times 10^{-4}$). Grating stimuli differed from spots in contrast as well as in motion, and both of these contributed to the difference in the observed strength of surround suppression (Supplementary Note 7 and Supplementary Fig. 10c, d). Thus, while confirming that VGluT3 cells have strong receptive-field surrounds, we find that their strength varies with stimulus configuration and that they do not preclude robust responses to global retinal motion.

### VGluT3 dendrites are tuned to the direction of motion

To our surprise, VGluT3 dendritic Ca$^{2+}$ signals were tuned for the direction of grating motion (Fig. 6). When presented with fast-moving full-field gratings (speed 1520 or 1910 µm/s), individual ROIs typically exhibited a preference for grating motion in either direction along a single axis (Fig. 6a–c); 84% of ROIs met the standard criterion for this form of orientation selectivity (OS; 206 of 245 ROIs, 11 FOVs, 6 mice; for the entire sample, OSI = 0.38 ± 0.01, $n$ = 245). On the other hand, many VGluT3 ROIs preferred motion in one direction; nearly two thirds of them (65%) met the standard criterion for direction selectivity (DS, 157 ROIs; entire sample DSI = 0.28 ± 0.01). Only 6% (15 ROIs) showed neither orientation nor direction preference. Over half (54%) met both criteria. In some FOVs, ROIs were either mostly orientation selective (Fig. 6a) or mostly direction selective (Fig. 6b, left), while other FOVs contained a mix of OS and DS preference (Fig. 6b, right). Overall most OS ROIs preferred an axis of motion oriented 30° or 210° relative to the nasotemporal axis. A smaller group of ROIs exhibited preferences clustered near the dorsoventral axis (80° or 260°) (Fig. 6d, mean overall preferred orientation =56 ± 3°, $n$ = 206). Preferred directions of DS ROIs clustered mainly near one of the two directions along the same axis as the main cluster of preferred orientations (Fig. 6e, 198 ± 6°, $n$ = 157). Note that we defined the preferred orientation as the preferred axis of motion (not e.g., as the orientation of the bars in the grating).

Surprisingly, the direction and orientation selectivity we observed in VGluT3 dendritic calcium responses was not clearly evident in the glycinergic inhibitory current in ON DSGCs under pharmacological manipulation (Supplementary Note 8 and Supplementary Fig. 11).

## Discussion

We have identified the inhibitory synaptic circuit primarily responsible for speed selectivity in ON DSGCs in mice, and therefore in OKN (but see ref. 35). It features a familiar amacrine cell type—the VGluT3 cell. We show that VGluT3 cells respond well to fast global motion, and that

they veto ON DSGC responses to such motion through feedforward glycinergic inhibition.

These conclusions are supported by several lines of convergent evidence. Our pharmacological findings recapitulate earlier evidence in rabbit implicating glycinergic inhibition as the key suppressor of fast motion responses in ON DSGCs. Connectomic analysis shows that just three types of amacrine cells account for nearly all conventional (non-ribbon) synaptic contacts onto ON DSGCs. Among these cell types, only the VGluT3 cell is glycinergic. We show that the VGluT3 plexus is strongly activated by rapid motion, and that this is true even during global retinal slip, despite the strong suppressive surrounds of VGluT3 receptive fields. We demonstrate that optogenetic activation of VGluT3 cells evokes glycinergic inhibitory currents in ON DSGCs whereas chemogenetic suppression of VGluT3 output reduces the inhibition of ON DSGCs triggered by rapid motion. The role of glycinergic inhibition in suppressing ON-DSGCs responses at high speeds has recently been confirmed in mouse[36]. Though our data implicate feedforward glycinergic inhibition from VGluT3 cells as the main determinant of slow speed preference in ON DSGCs, their speed-tuning profile is undoubtedly shaped to some extent also by their excitatory inputs[16] and by their GABAergic inputs from SACs and widefield amacrine cells.

The output of ON DSGCs to the accessory optic system encodes both components of the local vector of global retinal slip: direction and speed. The directional coding derives mainly from a specific amacrine-cell type, the SAC, through spatially asymmetric feedforward GABAergic inhibition of the ON DSGC. Here, we argue that VGluT3 cells specify the other vector component—speed. These dual-transmitter interneurons excite some RGCs through glutamate release, while inhibiting others through glycine. Two earlier studies reported only excitatory glutamatergic transmission from VGluT3 cells to ON DSGCs[17], not glycinergic inhibition[37]. We confirmed weak VGluT3-mediated excitation of ON DSGCs, but inhibition was much stronger. The discrepancy from the earlier work may be traceable our 20-fold dimmer optogenetic stimulus. As far as we are aware, this is the first example of any neuronal type receiving both excitatory and inhibitory

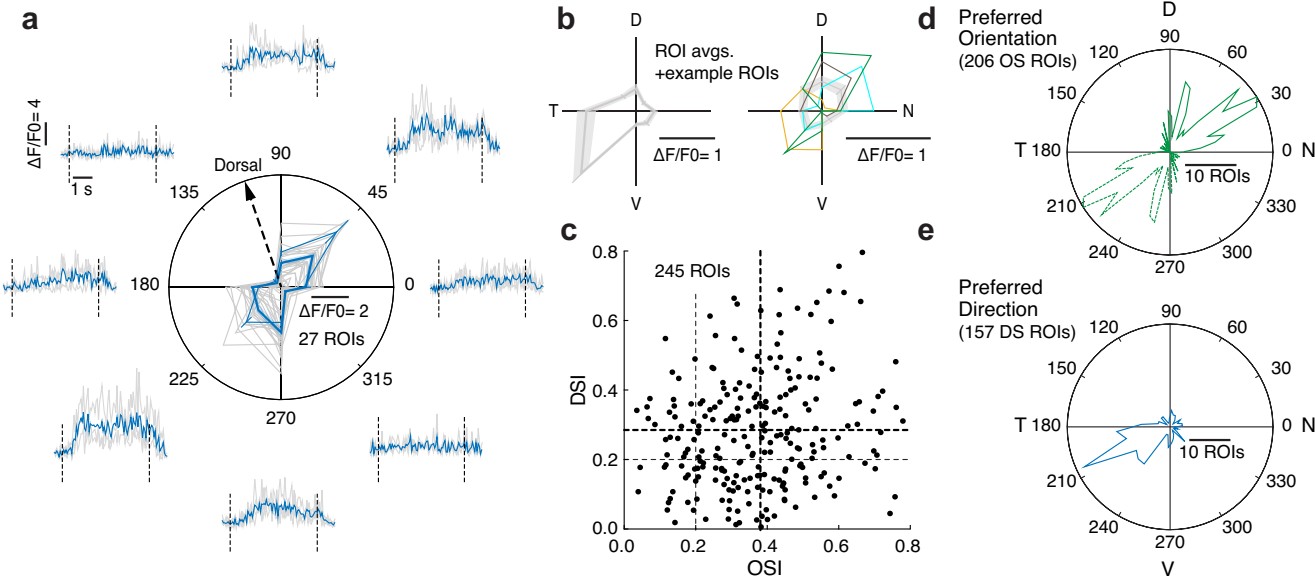

**Fig. 6 | Direction and orientation selectivity of VGluT3 dendrites. a** Calcium responses (ΔF/F, time averaged) in VGluT3 dendrites vs. direction of motion of a full-field grating (speed 1520 µm/s), for 27 ROIs (gray) and their mean ± SEM (thick blue line) in a single FOV. Example traces are shown (blue for mean response over 3 repeated trials, gray for single trials), taken from a single ROI (thin blue line on polar plot, mean ± SEM over trials). An arrow shows the dorsal direction of the retina. **b** Mean ± SEM responses in two more FOVs (Left: 16 ROIs, Right: 20 ROIs;

speed 1520 µm/s). Colored curves: example individual ROIs. **c** Orientation and direction selectivity indices (OSI/DSI) of 245 ROIs from 6 FOVs. Thick lines show distribution averages. A ROI was considered direction or orientation selective (OS/DS) if its OSI or DSI was ≥0.2, respectively (thin dashed lines). **d, e** Polar histograms of the preferred orientations (**d**) or preferred directions of grating motion (**e**) of OS or DS ROIs (206 OS and 157 DS ROIs from 6 FOVs). Source data are provided as a Source data file.

input from VGluT3 cells. This is reminiscent of the mixed excitatory and inhibitory inputs from SACs to DSGCs[38]. Our reconstructions reveal that SAC and VGluT3 inputs account for more than 92% of the total inhibitory synapses onto ON DSGCs. Both SACs and VGluT3 cells have been found in higher primate retinas[8,39–41], so the same two cell types may encode the retinal slip vector in the human retina. Genetic defects perturbing this circuit disrupt normal image stabilization in human patients[3].

It has been known for decades that ON DSGCs prefer slow speeds while ON-OFF DSGCs prefer faster speeds[9]. The mechanistic basis of this difference remained obscure until work in rabbit showed that only the ON DSGCs receive fast-speed glycinergic inhibition[10]. Here we confirm this in mice and identify VGluT3 cells as the source of inhibition. We also find that VGluT3 cells further differentiate the speed preferences of the two DSGC classes by preferentially augmenting fast-motion responses in ON-OFF DSGCs though excitatory glutamatergic synapses.

Are VGluT3 cells the sole source of glycinergic inhibition of ON DSGCs? Selective chemogenetic suppression of VGluT3 cells was less effective in eliminating feedforward glycinergic inhibition in ON DSGCs than global blockade of glycinergic inhibition with strychnine. This may indicate that there are other sources of glycinergic input. The connectomic analysis did identify two synapses from H18 amacrine cells[30,42] to ON DSGCs. If H18 cells are glycinergic, as suggested by their relatively small, highly branched dendritic arbors, their contributions to ON DSGC inhibition would have been blocked by strychnine but not by our chemogenetic manipulation. However, the transmitter composition of H18 cells is unknown, and they synapse onto ON DSGCs only very rarely. We therefore favor an alternate explanation—that VGluT3 cells are virtually the sole glycinergic input, but that we were only partly successful in chemogenetically blocking that input. This could have occurred because weak DREADD expression or biophysical factors did not allow complete chemogenetic suppression of VGluT3 glycine release. Partial suppression of VGluT3 cells could account for the finding that the CNO-sensitive inhibition was a fixed fraction of the total inhibition across speeds, and that the shapes of the inhibitory current traces were preserved following CNO application (Fig. 5a, b, Supplementary Note 6, Supplementary Fig. 8). These two models are not exhaustive; others mechanisms might account for the discrepancy between the chemogenetic and strychnine effects, such as indirect network effects of strychnine application.

In the presence of glycinergic blockade, optogenetic activation of VGlut3 cells evoked a presumptive GABA-mediated inhibitory current in some ON DSGCs. This suggests that VGluT3 cells supply glutamatergic excitation to one or both of the GABAergic amacrine cell types that synapse onto ON DSGCs, namely SACs and widefield ACs). However, SACs do not respond to optogenetic activation of VGluT3 cells[17], and our SBEM analysis confirms that there are very few synaptic contacts from VGluT3 cells to SACs. This leaves widefield amacrine cells as the likely conduit for this influence. Whatever the circuit responsible, this indirect influence of VGluT3 cells has a net inhibitory impact on ON DSGCs, just as the direct glycinergic influence does.

The connectomic evidence suggests that the direct influences of VGluT3 cells extend far more widely to other retinal neurons than appreciated in earlier functional surveys (Supplementary Note 2, Supplementary Table 1). They encompass diverse RGCs and amacrine cells as well as some cone bipolar cells. VGluT3 cells rarely synapse on other VGluT3 cells, a marked contrast to the extensive mutual inhibition between SACs.

Since SBEM analysis implied that virtually all types of RGCs receive some VGluT3 input, the role identified here for VGluT3 cells in sculpting the speed-tuning profiles of DSGCs (both ON and ON-OFF) is likely to extend broadly to the retinal output. For RGC types getting excitatory glutamatergic drive from VGluT3 cells, we would expect boosted sensitivity to relatively fast motion. OFF transient alpha cells

are one such type and our data confirm their high-speed sensitivity. When the VGluT3 contribution is inhibitory, we expect it to favor responses to slow speeds in the postsynaptic ganglion cells, as shown here for ON DSGCs. Just such a functional influence seems to be detectable in a bistratified medium-field RGC type variously termed ON-delayed[22], Suppressed by Contrast[43–45], R-cell[46], or Type 73[6]. They are ON-type RGCs, but their spiking response to light steps exhibits a characteristic delay derived in part from fast feedforward glycinergic inhibition[22]. Ostensibly the same type was inhibited by optogenetic activation of VGluT3 cells[44], and we confirm by SBEM that VGluT3 cells make synapses onto them. Taken together, these findings imply that this cell type might, like the ON DSGCs, prefer slow speeds. Our survey of speed tuning of various RGC types appears to confirm this, since ON-delayed cells were the only RGC type to prefer speeds as slow as those preferred by ON DSGCs (Fig. 1d).

$Ca^{2+}$ imaging revealed robust responses to global motion in VGluT3 dendrites. This is unexpected because VGluT3 cells are known to be strongly suppressed by visual stimulation of their receptive-field surrounds[17,33,34]. Responsiveness to global motion also seems at odds with the reported selectivity of these cells for local object motion[33,34]. We have shown that the strength of surround suppression is context dependent, as is often the case[47,48]. When the receptive field is probed with a moving grating of various sizes, rather than with flashed spots[17], the surround appears weaker and the center appears larger (Fig. 5g, h, Supplementary Fig. 10).

Though VGluT3 cells respond to global motion, they may nonetheless participate in object-motion sensing[33,34]. Indeed VGluT3 processes responded much more vigorously to small grating patches than to extended gratings (Fig. 5f). The previous studies demonstrating complete VGluT3 suppression during global motion used different stimulus parameters than ours: their gratings moved more slowly than optimal for VGluT3 activation (Fig. 5c–e). Further, gratings were presented much more briefly in the earlier studies (100 μm translation lasting 0.25–0.5 s) than in ours. In our hands, responses to full-field grating motion ramped up over ~2 s, and were barely detectable in the first second (e.g., Figure 5b), similarly to previous findings. This delayed VGluT3 $Ca^{2+}$ response contrasts with the brisk inhibitory currents evoked by the same stimuli in postsynaptic ON-DSGCs (Fig. 1c). In this context, it must be remembered that calcium signals could misrepresent true membrane voltage due to slow kinetics, nonlinearities, and insensitivity to near-threshold voltage changes. However, a fast contribution to inhibitory currents in ON-DSGCs from ON SACs cannot be ruled out.

If VGluT3 dendrites are selective for orientation or direction, can they still suppress fast-motion responses in all subtypes of ON DSGCs, which differ in their preferred directions? Because VGluT3 directional preference exhibited considerable variability, suppression of ON DSGCs should nonetheless occur for many directions despite the strong net preference. Indeed, we found speed tuning of inhibition in all ON DSGCs, regardless of their different preferred direction. Further, in single ON DSGCs we found no net directional or orientation bias in the inhibition evoked by fast speeds (Supplementary Note 8, Supplementary Fig. 11). Future work will be needed to understand why VGluT3 sensitivity to stimulus direction is not clearly reflected in their feedforward inhibition onto ON DSGCs.

## Methods
### Animals
All procedures were in accordance with the National Institutes of Health guidelines and approved by the Institutional Animal Care and Use Committee at Brown University. We studied adult mice of either sex, 2–8 months old. Mouse sex was not recorded. Mice were housed in a 12 light/12 dark cycle at room temperature (22 °C), 40–60% relative humidity. Wild-type mice were C57BL/6J (Jackson Laboratory). To target ON DSGCs for recording, we used HoxD10-GFP mice

(GENSAT collection, Tg(Hoxd10-EGFP)LT174Gsat/Mmucd, MMRRC #032065) and Pcdh9-Cre mice (GENSAT collection, Tg(Pcdh9-Cre)NP276Gsat/Mmucd, MMRRC #036084). The VGluT3-Cre line (The Jackson Laboratory, B6;129S-Slc17a8$^{tm1.1(cre)Hze}$/J, #028534) was crossed with each of four Cre-dependent mouse lines: Ai32 (Jackson, B6.Cg-Gt(ROSA)26Sor$^{tm32(CAG-COP4*H134R/EYFP)Hze}$/J, #024109) for optogenetics; DREADD mice (Jackson, B6.129-Gt(ROSA)26Sor$^{tm1(CAG-CHRM4*,-mCitrine)Ute}$/J, #026219) for chemogenetics, Ai14 (Jackson, B6;129S6-Gt(ROSA)26Sor$^{tm14(CAG-tdTomato)Hze}$/J, #007908) for fluorescent labeling for characterization of the VGluT3-Cre mouse, and Ai148 (Jackson, B6.Cg-Igs7$^{tm148.1(tetO-GCaMP6f,CAG-tTA2)Hze}$/J, #030328) for Ca$^{2+}$ imaging. For the Müller cell control experiment, GLAST-Cre mice (Jackson, Tg(Slc1a3-cre/ERT)1Nat/J, #012586) were crossed with Ai32.

### Retinal dissection

Isolation of the retina was performed similarly to ref. 49. Mice were euthanized by cervical dislocation. The eyes were removed and immersed in oxygenated Ames medium (95% O$_2$, 5% CO$_2$; Sigma-Aldrich; supplemented with 23 mM NaHCO$_3$ and 10 mM d-glucose). Under dim red light, the globe was incised, and the cornea, lens and vitreous humor were removed. A relieving ventral cut was made in the eyecup, and the retina was isolated. Three more cuts were made in the retina, roughly along the temporal, nasal and dorso-nasal directions. These were made asymmetrically to allow for disambiguation of retinal orientation. The retina was flat-mounted on a polylysine coverslip (Corning, #354086), which was secured in a recording chamber.

### Tamoxifen injections

Tamoxifen (Sigma-Aldrich) was dissolved in corn oil (Sigma-Aldrich; 20 mg/ml), sonicated (30 minutes, RT) and placed in hot water (2 h, 45 °C) and once homogenous, passed through a 0.2 μm filter. Tamoxifen was injected IP, 2–2.5 mg per mouse, once a day for 3 days. This resulted in dense YFP labeling of Müller glia in GLAST-Cre x Ai32 mice[50]. The animals were used in experiments three weeks after the last tamoxifen injection.

### Electrophysiology

Patch-clamp recordings of isolated flat-mount retina were performed under voltage-clamp using a Multiclamp 700B amplifier, Digidata 1550 digitizer, and pClamp 10.5 data acquisition software (Molecular Devices; 10 kHz sampling). Pipettes were pulled from thick-walled borosilicate tubing (P-97, Sutter Instruments). Retinas were continuously superfused during experiments with oxygenated Ames' medium at 32 °C, flow rate ~5 ml/min. For cell attached recordings, Ames filled pipettes were used (tip resistance of 4–5 MΩ). For whole-cell voltage clamp recordings, pipettes filled with cesium internal solution (In mM: Cs methane sulfonate, 104.7, TEA-Cl, 10, HEPES, 20, EGTA, 10, QX-314, 2, ATP-Mg, 5, GTP-Tris, 0.5, pH 7.3, osmolarity 276 mOsm; All purchased from Sigma-Aldrich) were used (tip resistance of 5.5–6.5 MΩ). To isolate excitatory and inhibitory synaptic currents, the recorded cell was held near the reversal potential for inhibition (~ −65 mV) and excitation (~ +20 mV), respectively. Application of synaptic blockers to the bath as well as Clozapine-N-Oxide (CNO) was done by switching the perfused medium into medium containing the blocker and waiting for ~7 minutes. The blockers used were: strychnine (1 μM, Sigma), SR95531 (10 μM, Sigma), L-AP4 (20 μM, Tocris), ACET (10 μM, Tocris), Hexamethonium (100 μM, Sigma), CNQX (20 μM, Tocris), D-AP5 (50 μM, Tocris). In the DREADD experiments, CNO was used to activate the DREADD (1 nM, Sigma).

### Light stimulation

Light stimuli were generated as in refs. 46,49. Patterned visual stimuli, synthesized by custom software using Psychophysics Toolbox under Matlab (The MathWorks), were projected (AX325AA, HP) and focused onto the photoreceptor outer segments through the microscope's condenser. The projected display covered ~1.5 × 1.5 mm (5.8 μm/pixel). The video projector was modified to use a single UV LED lamp (NC4U134A, Nichia). The LED's spectrum (385 ± 5 nm) shifted to a 395 ± 12 nm peak skewed towards shorter waelengths after transmission through the projector and condenser optics, as well as a 440 nm band-pass filter (FF01-440/SP, Semrock), and various reflective neutral density filters (Edmund Optics), and reflection at a dichroic mirror (T425lpxr, Chroma). The photoisomerization rates used were 2–5 × 10$^3$ R*/rod/s (1–2 × 10$^{-2}$ W·m$^{-2}$), and for the spectrum of the stimulus were similar among rods, M-cones and S-cones (see ref. 46 and references therein). The ratios between photoizomerization rates for rods, M- and S-cones depend solely on the spectrum of the light source, and were 2.5: 2: 1.7, respectively. In the beginning of a stimulus sequence, a uniform screen with the stimulus' mean intensity (gray) was projected for 20–30 s for light adaptation. To identify RGC types, the spike responses of the cell to a 460 μm diameter spot of +0.95 contrast were recorded. To assess the directional tuning of ON DSGCs we used full-field sinusoidal gratings (cycle = 377 μm, contrast = 0.95, stimulus duration = 5 s, inter-stimulus duration = 3 s at uniform mean grating intensity) drifting in 8 directions in a randomized sequence (drift speed = 226 μm/s, 4 repetitions). For speed response curves, the same grating was drifted in the preferred direction (DSGCs), at 7–8 different speeds at a randomized sequence, with 3 repetitions for each speed. For VGluT3 dendrites Ca$^{2+}$ imaging, the same stimulus was used, with the direction of motion chosen arbitrarily. The same was also presented in a circular patch on a gray background in Fig. 5f. In Fig. 5g and Supplementary Fig. 10a, b, white spots of different sizes were presented from dark, and the same grating stimulus as before was presented over gray in circular masks of different sizes. In Supplementary Fig. 10c, d, gratings and spots of different sizes (gray = the mean intensity of the gratings) were both presented from dark, and spots were gray rather than white (0.5 intensity in Supplementary Fig. 10c, d, relative to Fig. 5g, h and Supplementary Fig. 10a, b) to equalize the global contrast of the two stimuli. To assess the directional tuning of ROIs (Fig. 6) we used full-field gratings drifting in 8 directions as above, with speeds 1520 or 1900 μm/s. In all drifting gratings experiments, a grating with the same spatial frequency was used (377 μm cycle).

### Electrophysiological data analysis

Throughout the text, we list group data as mean ± standard error of the mean, unless otherwise specified. All data analysis was done using custom written Matlab procedures. Individual peri-stimulus time histograms (PSTH) presented for spikes, or current traces for voltage clamp recordings, were averaged over three trials. For speed tuning curves (e.g., Fig. 1a), the response was the total number of spikes (cell attached recordings) or the charge transfer (voltage clamp) over the stimulus presentation duration. In voltage clamp, the average baseline current, recorded for 0.5 s immediately before the stimulus, was subtracted from the current during the stimulus, which was then integrated to produce the charge transfer. In population response vs. speed data, curves from different cells were normalized by their maximum and averaged. If different cells in the population had data points at different speeds, their interpolated curves were averaged. A data point was marked if it was a real measured point in at least one cell. The 'optimal' speed for a cell's response curve was the speed at which the curve averaged over trials was maximal. The 'half maximum speed' was the speed at which the descending branch of the curve, as determined by linear interpolation between data points, crossed the horizontal line of half the maximal response, and was considered a cutoff speed for the cell's responses. In population data where synaptic blockers or CNO were used (DREADDs), the response curves for each cell were normalized by the maximum of the control curve for that cell, and then curves were averaged over cells. A change in the speed range of the responses (Fig. 1g) was measured as the speed difference

between the two curves at a response value of 0.5, where the maximal response at the control condition had been set equal to 1. The currents presented for optogenetics (Fig. 3) were recorded at 20 kHz. We subtracted a baseline current that was the average current in a 2 s prestimulus interval. The current traces were smoothed by averaging in 10 ms windows. Current traces in Fig. 3 were averaged over 3–5 trials. Maximum currents in these curves were summarized in the data of Fig. 3b, c, e, h, i. The noise level for each cell (and each holding voltage) was taken as the standard deviation of the smoothed current trace during the pre-stimulus interval, multiplied by 2. In Fig. 3b, c, e, h, i, this noise level was averaged over cells (denoted by a dotted line). The shaded regions in Fig. 3b, c, e, h denote the resulting average ± SD. In Fig. 3c, e, i, currents and noise levels were normalized by the maximal current in the control condition. Statistical significance was evaluated using the one-tailed paired Student t-test (Figs. 1e, g and 4a–c, f, g) and is shown for effects mentioned in the text, as well as for the differences at specific speeds, denoted by stars above or below error bars. 0, 1, 2, and 3 stars correspond to $p > 0.05$, $p < 0.05$, $p < 0.01$ and $p < 0.001$, respectively.

## Immunohistochemistry

Retinas were fixed and counterstained with the following antibodies. Primary antibodies: Goat anti-ChAT (Choline acetyltransferase; 1:200, Millipore Sigma #AB144); Rabbit anti-VGluT3 (1:250, Invitrogen #PA5-85784). Chicken anti-GFP (1:1000, Abcam #ab13970) was used to enhance the fluorescence of the Cre-dependent GFP virus. Rabbit anti-HA tag (1: 200, Cell Signaling Technology #3724) was used to stain the HA-tagged hM4Di receptor in the VGluT3 x DREADD mouse. Secondary antibodies: Donkey anti-Chicken 488 (1:1000, Jackson Immunoresearch #703-545-155); Donkey anti-Chicken 594 (1:1000, Jackson Immunoresearch #703-585-155); Donkey anti-Goat 488 (1:200, Invitrogen #A-11055); Donkey anti-Goat 594 (1:200, Invitrogen #A-11058); Donkey anti-Rabbit 647 (1:200, Invitrogen #A31573).

## Imaging for cell targeting and dendritic morphology

To target fluorescent cells for patch recording, two photon imaging was used (Olympus FV1200MPE BASIC (BX-61WI) microscope, 25×, 1.05 NA water-immersion objective (XLPL25XWMP, Olympus), an ultrafast pulsed laser (Mai Tai DeepSee HP, Spectra-Physics) tuned to 910 nm, and the imaging software Fluoview 4.1 (Olympus)). To acquire an image stack, RGCs were filled during electrophysiological recordings with Alexa hydrazide 488 or 594 (100 μM, Invitrogen), and were imaged following the recording, either using the two-photon or the single-photon (confocal) configuration of the two-photon microscope. Tissue in which fluorescent proteins were expressed was often fixed and immunostained (see above), and subsequently imaged on a confocal microscope (Olympus FV3000, UPlan Super Apochromat objectives, 30xS, 1.05 NA, or 60x2S, 1.3 NA) in the Leduc Imaging Facility, Brown University. Confocal and two-photon stacks were processed in Fiji (https://imagej.net/software/fiji), and collapsed using either maximum intensity or maximum standard deviation projections.

## Functional imaging

Imaging of calcium indicator signals were acquired using the two-photon microscope and conditions described above, as has been done previously[49]. The laser power used (910 nm) was ~3%. The frame rate was 15 Hz. For imaging responses in dendrites, 128 × 256 pixel fields of view were used with a zoom of 4.5× or 7× ($56 × 113$ or $36 × 73$ μm$^2$ FOVs). Light stimulus presentation was synchronized to the fly-back times in the scanning of the microscope so that they did not interfere with the measured signal.

## Functional imaging data analysis

Functional imaging analysis was done using Fiji (see above) and custom written Matlab routines. For each movie, a standard deviation

projection was made, over which ROIs were manually marked over brighter dendrites (Fig. 5a), and traces of their area-averaged brightness over time were acquired. For the baseline fluorescence $F_0$, we averaged the brightness over 0.5 s before every stimulus presentation. ROIs were chosen for analysis if their time averaged responses surpassed a threshold $\Delta F/F_0$ (0.3–0.6) during at least 3 or 6 stimulus presentations out of 24. ROIs that had exceptionally noisy responses, or that seemed in the projection image to belong to RGC dendrites rather than the VGluT3 plexus were discarded. In Figs. 5c, d and 6a, b, responses from responsive ROIs were averaged in a single FOV. In Fig. 5e, f, g, the ROI average curves from several FOVs were averaged over FOVs. In Fig. 5e, g, the curve from each FOV was normalized by their maximum. In Fig. 5f, both the masked and full-field grating response curves were normalized by the maximum of the masked grating curve. Time fluorescence traces in Figs. 5b, h and 6a, Supplementary Figs. 9 and 10 were taken from single example ROIs. Direction and orientation selectivity indices (DSI, OSI) and preferred directions and orientations were calculated from vector sums of responses in different directions of motion[49,51], and ROIs were considered orientation or direction selective if the corresponding index exceeded 0.2. In Fig. 6d, e, the curves are polar histograms summarizing the preferred orientations or directions of all cells that were OS or DS, respectively. Statistical significance in Fig. 5f, g was evaluated using the one-tailed, paired t-test for the set of FOVs under two different stimuli. Test results mentioned in the text and at specific speeds (Fig. 5f) or diameters (Fig. 5g) are shown as in previous figures. For ROI responses within FOVs (Fig. 5g), an unpaired, one-tailed t-test was used, as ROIs were chosen independently for each movie (This test was repeated in two FOVs).

## Intraocular injections

Mice were anaesthetized with isoflurane (3% in oxygen; Matrx VIP 3000, Midmark). A viral vector inducing Cre-dependent expression of a payload (see below) was injected into the vitreous humor of the right eye through a glass pipette using a microinjector (Picospritzer III, Science Products GmbH). Analgesia (Proparacaine, eye drops) was applied to the eye ~2 min before the injection, and immediately following the injection (Buprenorphine SR, 0.02 ml, intraperitoneal) to minimize postoperative pain. Mice were then taken off anesthesia, recovered within several minutes, and monitored for 48 h following the procedure. Animals were killed and retinas removed 14–21 d later.

## Viruses

pAAV2/2-hSyn-DIO-hM4D(Gi)-mCherry (Addgene #44362, Roth Lab) was injected intraocularly in VGluT3-Cre mice, causing Cre-dependent expression of hM4D(Gi), a modified human muscarinic M4 receptor, that is an inhibitory Designer Receptor Exclusively Activated by Designer Drugs (DREADD). AAV2/2-EF1a-DIO-hChR2(H134R)-EYFP (from UNC vector core, Deisseroth Lab) was injected VGluT3-Cre mice to express ChR2 in a Cre-dependent manner, but mostly caused expression in too few of the VGluT3 ACs to drive optogenetic responses in postsynaptic RGCs (Supplementary Fig. 4d). rAAV2/2-CAG-flex-GFP (UNC vector core) was injected in Pcdh9-Cre mice to target ON DSGCs for recording.

## Optogenetics

Light stimulation to activate ChR2 was generated using a LED light source (Mightex MLS-5500-MK1; LED driver: open-ephys.org, Cyclops) and introduced through the microscope objective and GFP excitation filter cube, resulting in a spectrum peak at 480 ± 10 nm, and illumination over an area of 1 mm in diameter. The light intensity at the sample was 0.9–1.4 nW/μm$^2$, the lowest intensity that yielded robust postsynaptic responses in ON DSGCs. The stimulation time was 0.1 s or 1 s for 5 repeats, 4 s between repeats. The intensity and time were optimized for a robust response, while minimizing the rundown of the

response over time, and the driving of large bursts of current that were sometimes observed, that were inconsistent over trials or not stimulus-locked. Cells exhibiting such currents were excluded from the data.

## Electron microscopy neuronal reconstructions

An existing dataset of retinal sections from a Serial Blockface Electron Microscope (SBEM), 'k0725'[25] was used. The imaged volume dimensions were $50 \times 210 \times 260$ μm$^3$ with the short dimension spanning the IPL and parts of the GCL and INL layers of the retina. The pixel size was 13.2 nm$^2$ and the section thickness 26 nm. The images contained intracellular details, e.g., synaptic vesicles. The webKnossos platform (v. 0.12.3; https://webknossos.org[52]) was used for tracing neuronal skeletons and annotating synapses. For the plot of stratification depth (Supplementary Fig. 3j), the depth of nodes in the skeletons were normalized to the depth of the ON SAC plexus to correct for the tilt and curvature of the tissue in the block[53].

## Statistics and reproducibility

Averaging, error estimation and the statistical tests used are detailed in the "Methods" sections above. No adjustments were made to p values. Independent repetitions: each cell in the following data was recorded in a different retina and mouse: Figs. 1a, e, g, 3 and 4a–c, f, g. Supplementary Figs. 2, 5–8. Numbers of cells: Figs. 1a, 12, 1e, 6, 1g, 4. Figure 1d, 60 mice (ON DSGCs from 49 mice). Figure 3a–e, g–i, all from 13 cells. Figure 4a–c, f, g, 3 cells for each. Supplementary Fig. 2c, 2, Supplementary Fig. 8c, d, 3 each. In Figs. 5, 6, Supplementary Figs. 9, 10, several FOVs were recorded in one retina per mouse. Numbers of mice: Fig. 5c–e, 4, Fig. 5f, 2, 5g, 2. Figure 6a, b, 2. Figure 6c–f, 6. Supplementary Fig. 10d, 2 mice. In Fig. 2, the analysis was done for a single ON DSGC. Characterization of the VGluT3-cre line: Supplementary Fig. 4a +e: 9 stacks, 2 retinas, single mouse (VGluT3-Ai14). Also performed in 23 stacks, 3 retinas, 2 mice (VGluT3-Cre + Cre-dependent GFP injected virus). Supplementary Fig. 4b: 8 stacks, one retina. Supplementary Fig. 4c: 6 stacks, one retina. Supplementary Fig. 4d: 7 stacks, 4 retinas, 4 mice.

## Reporting summary

Further information on research design is available in the Nature Portfolio Reporting Summary linked to this article.

## Data availability

Single cell recordings and calcium imaging data are available at: https://doi.org/10.5281/zenodo.7947135[54]. Electron-microscopic reconstructions are available at knossos.org/links/EBfAuezVRmNWiAt4. The k0725 SBEM dataset[25] is available at: https://webknossos.org/datasets/Demo_Organization/110629_k0725#2496,8000,5056,0,1.3. Source data are provided with this paper.

## Code availability

Custom written code used for analysis of electrophysiology and functional imaging data are available at: https://doi.org/10.5281/zenodo.7947135[54].

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

## Acknowledgements
The authors are grateful to Roman Drabchuk for help in intraocular injections and SBEM cell tracing, to Rachel Gunderson for careful annotation of the $Ca^{2+}$ imaging movies, to Dianne Boghossian for technical support, and to Erin Edwards. Supported by NIH grant R01EY012793 to D.M.B.

## Author contributions
A.M. designed the study, conducted experiments, analyzed the data and wrote the paper. X.Y. analyzed SBEM data, characterized the VGluT3-cre mouse and performed intraocular injections. T.A.Z. analyzed SBEM data and performed intraocular injections. M.L.L. and D.S. analyzed SBEM data. D.M.B. conceptualized and designed the study, wrote the paper, and analyzed SBEM data.

## Competing interests
The authors declare no competing interests.
