## [Peer Review File · Nature Communications]

REVIEWER COMMENTS

Reviewer #1 (Remarks to the Author):

In this work, Mani and colleagues demonstrate a vital and previously unappreciated role for the vGluT3 amacrine cell in setting the speed tuning of On-type direction-selective ganglion cells. The experiments are very well designed and executed and the writing is clear. Further, the findings should be of interest to a general audience. I have only a few minor comments that the authors might find useful.

1) Throughout the text, the authors reference changes in inhibitory synaptic input, etc with the appropriate number of cells in each condition. However, there is a general lack of statistical tests and accompanying text to orient the reader about how significant these changes are. I would suggest that the authors add this to the text where appropriate.

2) In Figure 1, the authors present spike, excitatory, and inhibitory synaptic currents. The plots are clear and summarize the effect nicely. One piece was unclear to me, however. Even at high speeds (e.g., 1796 $\mu\text{m/s}$), there is significant excitatory drive. Generally, the rule of thumb is that inhibition needs to be ~ 2 -3 times larger than excitation to suppress spiking given the differences in driving force near spike threshold, which is why I'm a bit surprised that spiking wasn't observed at that speed, particularly at stimulus onset. The story would change if On DS cells had a higher spike threshold. Is this the case? Do the authors think that intrinsic properties such as spike generation might come into play here? (i.e., Increasing the speed equates to increasing the temporal frequency for gratings; thus, if the membrane properties of the cell can't follow at high frequencies, this could also account for part of the effect).

3) I find Figure 1C a bit difficult to read because it is showing several cell types on the same count histogram. For example, it's not clear to me that some of the cell counts aren't hidden behind bars... One solution would be to plot probability on the y-axis instead of count and show the cell types as solid lines instead of bars. The counts could then go in the legend next to the cell type labels. I think this would clarify the graph quite a bit.

4) On line 385. "Evoked EPSCs were detected in only two of the 16 cells..." I don't know what *detected* means here? Generally this could be quantified in some way. For example, citing the mean/SEM of currents or charge across the 16 cells. This would then lend itself to a more formal statistical analysis.

Reviewer #2 (Remarks to the Author):

Image-stabilization is ubiquitous in visual systems and relies on rapid, reflexive detection of retinal slip to generate compensatory eye movements. The speed and direction of retinal slip is detected by specific retinal ganglion cells, the so-called ON-direction-selective ganglion cells (ON DSGCs). These neurons detect slow movement during slip without responding to rapid motion, which would countermand necessary ballistic eye-movements. Glycinergic pathways were implicated previously in the speed-tuning of these neurons. This study provides further support for those conclusions and identifies the VGLuT3 amacrine cells (ACs) as a potential source of the glycinergic inhibition. Most intriguingly, VGLuT3 ACs are both excitatory (glutamatergic) and inhibitory (glycinergic). Previous work showed that VGLuT3 ACs enhance responses of postsynaptic ganglion cells to high velocities via glutamatergic connections. The results presented here suggest that their glycinergic output may serve a complementary role by suppressing responses to high velocities and thus tuning postsynaptic ON DSGCs to respond preferentially to low speeds.

This work is likely to have significant impact because it further elucidates the connectivity and function of inhibitory interneurons in the retina. It should be of broad interest to the neuroscience community since much remains to be learned about the properties of inhibitory interneurons and their roles in neural signal processing. Importantly, the findings diverge from previous analyses of VGlut3 AC function, which is important, as it will hopefully prompt further reevaluation of the functional properties of these neurons.

Comments : those marked by asterisks are seen as more important.

1. ** The paper provides a range of evidence and builds a strong case that the VGlut3 ACs provide glycinergic input to On DSGCs and are thus well-placed to establish the slow speed tuning in these cells. The authors' central conclusion is that they have identified "...the inhibitory synaptic circuit primarily responsible for this speed selectivity in mice" (lines 564-568). I agree that the evidence is very strong, but there is no data showing that On DSGC speed tuning in the mouse is actually changed in the absence of glycinergic input. Such data might look like Fig. 1A, showing spiking responses or EPSPs before and after the addition of strychnine, or a similar experiment in the DREAD model before and after application of CNO. As it stands, I would be very surprised if such experiments didn't turn out as expected, but there is no data that directly confirms this expectation. Perhaps there's a good rationale for this omission?
2. Line 34-35: Image stabilization circuitry is also well-established in invertebrate systems.
3. ** Apart from showing SEMs, there is an almost complete lack of any statistical analysis. For each conclusion reached, the authors should document the statistical significance of the supporting data.
4. Many plots show averaged data normalized to the maximum, which is fine, but amplitude information is lost. It would be informative to quote the corresponding average maximum amplitudes in the associated figure legends. This would give the reader an idea, for example, the differences in the magnitudes of the inhibitory and the excitatory inputs in Fig. 1A.
5. ** Figs. 1A,D, 5A,B,F,G & 6C-G : please show the data-points and error bars in these figure panels rather than just the lines connecting the data-points. Visually, the images are dominated by the linear interpolation between the data-points rather than the actual data. It is very difficult to see what stimulus speeds were sampled in each plot.
6. Fig. 6C,D: Are there data-points missing on the right?
7. Line 138: 25% of cells responded best to slowest speed. Is it possible there is more than one type of On DSGC in the mouse as had been reported in the rabbit?
8. ** Lines 141, 144, 145, 157, 158, 163: Estimates and SEMs of the optimal speeds for different parameters are presented. Please explain how these values were calculated from the various tuning curves.
9. ** Line 186: Is this change statistically significant?
10. Fig.4A,F. Unless there is a compelling rationale for displaying "raw" traces, I would suggest filtering them to remove high-frequency noise, as in other panels, so that the time-courses of the evoked responses can be seen more clearly.
11. Lines 350-354: Perhaps I am missing the point, but the suggestion that the presence of sustained inhibition in the two experiments supports a role for VGlut3 ACs seems like a rather weak argument.
12. ** Fig. 5: For the most part, summary data is shown. It seems important to show some of the records that formed the basis for these results.

13. ** Lines 442-444: "The reduction was more pronounced at faster speeds..." Was this change in the speed tuning under the two conditions statistically significant?

14. ** Fig. 6A-F: Please describe the visual stimuli clearly here (dimension, spatial frequency, timing, etc), since they are essential to understand the data. These results appear to be at odds with previous work, as outlined here and in the Discussion, and therefore it seems important to illustrate primary records that underly the data in panels D,E,F.

15. ** Fig. 6G-I: The finding that calcium signals in VGlutT3 dendrites can be either orientation sensitive, direction sensitive, or both is interesting. Since this result seems to be rather novel and unexpected, primary data illustrating the effects should be shown. Did the glycinergic inputs to the On DSGCs show similar tuning?

16. Fig. 6G: I assume that the 8 stimulus directions were at 45 degree increments from 0 to 315. If so, the data in the top panel in Fig. 6G appears to be rotated somehow?

17. ** Lines 856-860: Please clarify the description of the light stimulus. What is meant by the "...LED's peak wavelength (385 nm) shifted to 395nm..."? How is it that longer wavelengths were preferentially transmitted through a short-pass dichroic filter? Please explain the stimulus parameters used to achieve similar photoisomerization rates in rods, M-cones and S-cones.

18. Lines 747-749; The authors note that "...the lack of a sudden and sustained contrast step led to a weaker surround suppression." I would have thought that the appearance of a grating stimulus in the current experiments would constitute a sudden and sustained contrast step?

19. Lines 757-759; Stimulus duration seems like an unlikely explanation for the differences observed. In any case, functional properties at short delays are probably most physiologically significant for detecting retinal slip.

Reviewer #3 (Remarks to the Author):

What are the noteworthy results? The authors investigate which influence the VGlut3 amacrine cells have on the response properties of direction selective ganglion cells in the mouse retina. The result is stunning and well supported by the data. VGlut3 cells inhibit the responses of On DS-cells to higher speeds via glycinergic action and augment the response of ON-Off Ds-cells to high speeds probably via glutamatergic action. More broadly, they suggest VGlut3 cells shape the response of many RGCs and amacrine cells to fast motion.

The work will be of significance to the field and related fields:

All experiments are done in-vitro on the explanted retina. It is thus a little astonishing how much emphasis the authors put on the well established (not from the authors of this study) classic role of On Ds-cells for image-stabilizing eye-movements. It would be enough to mention that once in the introduction. After reading the title and the first sentences in the introduction I was expecting experiment to show the behavioural effects of manipulating the retinal circuits on eye movements. But I was disappointed in this respect.

The work supports the conclusions and claims:

Nevertheless this is an impressive study on the retinal circuitry vetoing the response of ON DS-cells to higher velocities. However the visual system of many mammals goes to great effort to overcome this shortcoming of retinal ON DS-cells by having a visual cortex to provide responses to higher speeds. This may even be so in the mouse (but see Scanziani Nature 2016). Also it is not only the slip during VOR which is minimized by the OKR. Because the OKR is a closed loop system residual slip is indeed very slow.

The methodology is sound.

The electrophysiological experiments, including pharmacological and chemogenetic manipulation of the influence of the VGLUT3 amacrine cells on other retinal cells are state of the art. Unfortunately the outcome of the application of DREADD system is a little unconvincing and it does not really add important information to the results with the strychnine application. In my judgement the DREADD section could be omitted. The authors have a difficult time to explain the not so perfect results and leave the reader in a puzzle.

Also the electron microscopic reconstruction section interrupts the elegant flow of the physiological experiments and interpretation. Maybe it could be taken out here and serve as a building block for another more extensive morphological study and argumentation. I am absolutely not an expert on this topic and can give only a biased view.

These 2 omissions would focus the reader on the impressive physiological data and streamline the interpretation. I do not make this a condition sine qua non to highly recommend the ms for publication after minor review.

The work meets the expected standards in my field. I have to admit that I am not an expert in the methods but I can compliment on the results and the story.

With the above mentioned limitations for me there is enough detail provided in the methods for the work to be reproduced.

Specific comments:

Line 25: different motion speeds (instead of only fast)

59 there is a more direct link from the AOS to the oculomotor system through the nucleus prepositus hypoglossi

266 ff plexus plural is plexuses

218-328 aesthetically pleasing but not really important in the context of the physiological study

435 VGLUT3 cells excite ??

Discussion: cortical input shown by Scanziani. Monkeys and cats

647 an interneuron

406 + 579 + 654 + 664 Chemogenetic suppression is incomplete or only partly effective, could there be a speed-tuned excitatory input ??

606 slow retinal speed is also and probably mainly due to closed-loop mechanisms. OKR reduces speed even further

640 saccades are not influenced by OKR but the reflex may stabilize the eye after saccades (Fred Miles)

711 that that

779 4 cardinal directions does not apply to ON DS cells. What are the 4 ON DS subtypes ???

787 is there any evidence?

REVIEWER COMMENTS

We thank the reviewers for their positive feedback on our work and for their helpful suggestions for improvement.

Reviewer #1 (Remarks to the Author):

1) Throughout the text, the authors reference changes in inhibitory synaptic input, etc with the appropriate number of cells in each condition. However, there is a general lack of statistical tests and accompanying text to orient the reader about how significant these changes are. I would suggest that the authors add this to the text where appropriate.

This is an absolutely fair point. We have worked hard to address this in the revised manuscript. Statistical tests have been added in many places in the text and graphically in the figures. We performed additional experiments, adding more data in pursuit of statistical significance. The calcium imaging figure (Fig. 5) has been revised completely to clarify the sizes of the datasets and the statistics, and this figure now includes a dataset much larger than the previous one. Specifically - statistical tests have been added in the following panels: 1E, 1G (new panel), 3B, 3C, 3E, 3H, 3I – these five have all been added to clarify statistics, and include new analysis (see also comment 4 by the same reviewer). 4A-C, 4F, 4G (new panel), 5C-E (revised) 5F, 5G. Clarifications regarding statistical significance, sample sizes etc. were added throughout the Results sections pertaining to figures 1,3,4,5. In the following panels new data has been added to improve the statistical strength of the paper: 5E, 5F, 5G, 5K-M.

2) In Figure 1, the authors present spike, excitatory, and inhibitory synaptic currents. The plots are clear and summarize the effect nicely. One piece was unclear to me, however. Even at high speeds (e.g., 1796 $\mu\text{m/s}$), there is significant excitatory drive. Generally, the rule of thumb is that inhibition needs to be ~2-3 times larger than excitation to suppress spiking given the differences in driving force near spike threshold, which is why I'm a bit surprised that spiking wasn't observed at that speed, particularly at stimulus onset. The story would change if On DS cells had a higher spike threshold. Is this the case? Do the authors think that intrinsic properties such as spike generation might come into play here? (i.e., Increasing the speed equates to increasing the temporal frequency for gratings; thus, if the membrane properties of the cell can't follow at high frequencies, this could also account for part of the effect).

In response to this comment and others, we have added Supplementary Fig. 2A, 2B, and Fig. 1G, H. The text clarifying this has been added in lines 451-529 ("The ratio between..."), 614-617 ("In the presence of strychnine...")

We started by checking more carefully the E/I ratio in our data. The previous versions of Fig. 1A, B alone may have been a bit misleading in this context, as A shows curves normalized by their maxima, and B had a different

scale for excitation and inhibition. This is fixed in the revised figure. In addition, we plot the non-normalized version of 1A (Supplemental Fig. 2A). We further show the E/I ratio calculated for each cell and averaged over all cells, along with the spiking (Supplementary Fig. 2B). The suppression of spiking indeed fits the rule of thumb the reviewer mentions. Further, we now illustrate data showing that in the presence of strychnine ON DSGC spiking follows the high-frequency contrast modulation that occurs during fast grating drift (Fig. 1G,H). For all these reasons, we conclude that at least in this context it is principally feedforward inhibition rather than intrinsic membrane properties that limit the cell's response to fast motion.

3) I find Figure 1C a bit difficult to read because it is showing several cell types on the same count histogram. For example, it's not clear to me that some of the cell counts aren't hidden behind bars... One solution would be to plot probability on the y-axis instead of count and show the cell types as solid lines instead of bars. The counts could then go in the legend next to the cell type labels. I think this would clarify the graph quite a bit.

This is a great suggestion. We have implemented the solution recommended by the reviewer in the revised version of Fig. 1D.

4) On line 385. "Evoked EPSCs were detected in only two of the 16 cells..." I don't know what *detected* means here? Generally this could be quantified in some way. For example, citing the mean/SEM of currents or charge across the 16 cells. This would then lend itself to a more formal statistical analysis.

This section has been revised to include new analysis and much more statistical information. We replaced the "detected currents" phrasing with a more appropriate comparison of the response currents to the noise level (lines 1015-1017 "We detected optogenetically..."). We now show mean, SEM and the data distributions for the optogenetics data, in the text and Fig. 4B, C, E, H, and I. Statistical tests in the pharmacology section would not be very informative, due to the small sample size and the problem of current rundown over repetitions (Supplementary Fig. 6A and accompanying text).

Reviewer #2 (Remarks to the Author):

Comments : those marked by asterisks are seen as more important.

1. ** The paper provides a range of evidence and builds a strong case that the VGlutT3 ACs provide glycinergic input to On DSGCs and are thus well-placed to establish the slow speed tuning in these cells. The authors' central conclusion is that they have identified "...the inhibitory synaptic circuit primarily responsible for this speed selectivity in mice" (lines 564-568). I agree that the evidence is very strong, but there is no data showing that On DSGC speed tuning in the mouse is actually changed in the absence of glycinergic input. Such data might look like Fig. 1A, showing spiking responses or EPSPs before and after the addition of

strychnine, or a similar experiment in the DREAD model before and after application of CNO. As it stands, I would be very surprised if such experiments didn't turn out as expected, but there is no data that directly confirms this expectation. Perhaps there's a good rationale for this omission?

This is a good point. We conducted additional experiments to characterize ON-DSGC spiking before and after suppressing the glycinergic VGLuT3 inhibition. We confirmed that strychnine indeed extends the speed range that evokes ON DSGC spiking. This is summarized in two new panels, Fig. 1G, H, and in lines 610-618 ("Consistent with the large ..."). When VGLuT3 cells were suppressed with the DREADD ligand CNO we saw little augmentation of spiking at high speeds. We believe this is yet another indication of the incompleteness of DREADD suppression of VGLuT3 cells, and possibly even more so at the level of spikes, e.g. because of a spiking nonlinearity at the ganglion cell.

2. Line 34-35: Image stabilization circuitry is also well-established in invertebrate systems.

True, lines 99-100 were modified ("widespread among animals").

3. ** Apart from showing SEMs, there is an almost complete lack of any statistical analysis. For each conclusion reached, the authors should document the statistical significance of the supporting data.

This echoes Comment 1 from Reviewer #1. As described in detail in our response to that comment, we have worked hard to upgrade the statistical rigor, as reflected in revised figures and text.

4. Many plots show averaged data normalized to the maximum, which is fine, but amplitude information is lost. It would be informative to quote the corresponding average maximum amplitudes in the associated figure legends. This would give the reader an idea, for example, the differences in the magnitudes of the inhibitory and the excitatory inputs in Fig. 1A.

We agree, and have made this change to the text and figure legends, panels 1A, 4A, 4B, 4F, 4G, 5E-G. We have included a new analysis of the absolute excitation and inhibition from Fig. 1A, in Supplementary Fig. 2A, B.

5. ** Figs. 1A,D, 5A,B,F,G & 6C-G : please show the data-points and error bars in these figure panels rather than just the lines connecting the data-points. Visually, the images are dominated by the linear interpolation between the data-points rather than the actual data. It is very difficult to see what stimulus speeds were sampled in each plot.

Fixed, except in the panels 6G where the whole distribution of ROI curves is now shown in gray, in 5K which is a scatter plot, and in 5L,M that are polar histograms, containing the entire collections of 206 OS (top) or 157 DS (bottom) ROIs. In many panels, curves sampled at different points were grouped and averaged together (e.g. 1A, 4A). A data point and an error value are now provided where at least one of the curves in the population has a real data point, rather than an interpolated one.

6. Fig. 6C,D: Are there data-points missing on the right?

Fixed, and in other panels as well.

7. Line 138: 25% of cells responded best to slowest speed. Is it possible there is more than one type of On DSGC in the mouse as had been reported in the rabbit?

A comment about this was added in line 24 of the Supplementary Information (“The methods of...”).

8. ** Lines 141, 144, 145, 157, 158, 163: Estimates and SEMs of the optimal speeds for different parameters are presented. Please explain how these values were calculated from the various tuning curves.

An explanation was added in Methods, lines 2700-2712 (“The ‘optimal’ speed for...”). In line 391 where half max speeds are first used, we refer to Methods.

9. ** Line 186: Is this change statistically significant?

Yes. Statistical test added in Fig. 1E and in the text, line 547 (“inhibition was reduced by...”).

10. Fig.4A, F. Unless there is a compelling rationale for displaying “raw” traces, I would suggest filtering them to remove high-frequency noise, as in other panels, so that the time-courses of the evoked responses can be seen more clearly.

The previous Fig. 4 (now Fig. 3) contained both the raw and the filtered traces. We removed the raw traces as the reviewer suggested (they are now shown in Supplementary Fig. 5A, B).

11. Lines 350-354: Perhaps I am missing the point, but the suggestion that the presence of sustained inhibition in the two experiments supports a role for VGlut3 ACs seems like a rather weak argument.

Both the figure panel and the text have been moved to Supplementary Information (Supplementary Fig. 5C). The text the reviewer refers to has been changed to make the argument clearer, in lines 253-264 in Supplementary Information.

12. ** Fig. 5: For the most part, summary data is shown. It seems important to show some of the records that formed the basis for these results.

The records have been added in Supplementary Figure 7 along with the summary response curves in the same cells. (The original figure in the main text is now Fig. 4).

13. ** Lines 442-444: “The reduction was more pronounced at faster speeds...” Was this change in the speed tuning under the two conditions statistically significant?

Almost. See modified/added text in line 1342 (“The reduction appeared...”). Statistical tests added in figure. New data in figure 4G has been added (DREADD with spikes rather than excitation) where a similar parameter changed with significance.

14. ** Fig. 6A-F: Please describe the visual stimuli clearly here (dimension, spatial frequency, timing, etc.), since they are essential to understand the data.

Stimulus description was added: text, lines 1510-1513 (“To probe the spatial organization...”); Figure 5 legend and Supplementary Figs. 9, 10 legends. Stimulus schematics changed in panel 5H and shown in Supplementary Figs. 9, 10. More details added in Methods, lines 2680-2686 (“were both presented...”).

These results appear to be at odds with previous work, as outlined here and in the Discussion, and therefore it seems important to illustrate primary records that underly the data in panels D,E,F.

New Supplementary Figs. 9, 10, show fluorescence traces for the aforementioned data, along with summary plots for the specific FOVs. The previous panel 6F has been moved to the Supplementary Information altogether, now as Supplementary Fig. 10, panels C, D (area-response functions for spots and masked gratings with equal contrast).

15. ** Fig. 6G-I: The finding that calcium signals in VGlutT3 dendrites can be either orientation sensitive, direction sensitive, or both is interesting. Since this result seems to be rather novel and unexpected, primary data illustrating the effects should be shown.

Primary data are now shown in Fig. 5G.

Did the glycinergic inputs to the On DSGCs show similar tuning?

We conducted additional experiments to address this interesting question measuring inhibition in different directions, as well as during pharmacology manipulations, as summarized in Supplementary Fig. 11. In general, inhibitory currents in ON DSGC did not recapitulate the orientation/direction selectivity seen in VGlutT3 dendrites. We describe these experiments in the accompanying supplementary text (and briefly in line 1683, “Surprisingly, the direction...”, main text) and mention this seeming paradox in the Discussion.

16. Fig. 6G: I assume that the 8 stimulus directions were at 45 degree increments from 0 to 315. If so, the data in the top panel in Fig. 6G appears to be rotated somehow?

The original panel has been replaced with 5G and 5H in the revised figure. 5G is no longer rotated, and the dorsal retinal direction is denoted by an arrow. In panel 6H the orientation of the retina in the dish ensured that 0, 90°, 180° and 270° aligned with the nasal, dorsal, temporal and ventral directions respectively.

17. ** Lines 856-860: Please clarify the description of the light stimulus. What is

meant by the "...LED's peak wavelength (385 nm) shifted to 395nm..."? How is it that longer wavelengths were preferentially transmitted through a short-pass dichroic filter? Please explain the stimulus parameters used to achieve similar photoisomerization rates in rods, M-cones and S-cones.

The information in this paragraph has been verified and the paragraph revised, and now contains more details (lines 2642-2650 "LED lamp (NC4U134A, Nichia)... "). The filter is a band-pass with probably little effect on its own, but the stimulus is projected through several optical components that might be attenuating its short wavelength tail. The only parameter that the ratios between R^* , M^* and S^* depend on is the spectrum of the stimulus, see text.

18. Lines 747-749; The authors note that "...the lack of a sudden and sustained contrast step led to a weaker surround suppression." I would have thought that the appearance of a grating stimulus in the current experiments would constitute a sudden and sustained contrast step?

The artificial appearance of the stimulus indeed has a contribution to the response for both stimuli. This sentence has been deleted (line 2304, "During retinal slip of a natural...").

19. Lines 757-759; Stimulus duration seems like an unlikely explanation for the differences observed. In any case, functional properties at short delays are probably most physiologically significant for detecting retinal slip.

The paragraph has been changed to stress that the slower speeds used in the previous studies are not optimal for eliciting VGlut3 responses to full-field gratings. Those studies also used a short stimulus duration (0.25-0.5s), whereas ours was much longer. In fact, the full-field grating responses we evoked took time to develop, rather than beginning with the onset of motion (see Discussion, line 1875, "these previous studies... "and Fig. 5B). We agree that VGlut3 cells should be capable of rapidly detecting retinal slip both because a fast, transient, strychnine-sensitive inhibitory current is evoked shortly after motion onset (Fig. 1E bottom right panel) and because slip during saccades needs to suppress OKN quickly. We don't have a good explanation for the delayed calcium response in the VGlut3 cells, but suggest that the calcium signal may misrepresent the true membrane voltage due to slow kinetics, nonlinearities, and insensitivity to near-threshold voltage changes. The appearance of the grating in our study may have initially activated the surround more strongly as well.

Reviewer #3 (Remarks to the Author):

All experiments are done in-vitro on the explanted retina. It is thus a little astonishing how much emphasis the authors put on the well established (not from the authors of this study) classic role of On Ds-cells for image-stabilizing eye-movements. It would be enough to mention that once in the introduction. After reading the title and the first sentences in the introduction I was expecting

experiment to show the behavioral effects of manipulating the retinal circuits on eye movements. But I was disappointed in this respect.

We are certainly sorry to disappoint! It was not for lack of trying. We had always hoped to include a behavioral result with these findings and established a collaboration with another group (J.C. Cang, Univ. of Virginia) to make the necessary measurements. Unfortunately, we had to rely on the DREADD approach to suppress the VGlut3 contribution. Our physiological data strongly suggest that the suppression of VGlut3 activity we achieved was relatively modest, and any effect on OKN may have been obscured by the inherent variability in the behavioral data. Since we couldn't deliver the behavioral result, we have revised the text to reduce the emphasis on this behavioral dimension of our work. We still frame our results in the context of the well-established role for the fast-speed inhibition of ON DSGCs in the optokinetic behavior of mammals. However, we have changed the title in an effort to signal that our study is focused on the retinal level. In addition, we have removed information/comments regarding OKN that were not directly relevant to the results of the current study: line 12, 20 (abstract), lines 99 ("Animals see better..."), 1748 ("Optokinetic image stabilization is..."), 1867 ("Image stabilization is ubiquitous..."), 2047 ("The two classes....") in the introduction and discussion.

... this is an impressive study on the retinal circuitry vetoing the response of ON DS-cells to higher velocities. However the visual system of many mammals goes to great effort to overcome this shortcoming of retinal ON DS-cells by having a visual cortex to provide responses to higher speeds. This may even be so in the mouse (but see Scanziani Nature 2016). Also it is not only the slip during VOR which is minimized by the OKR. Because the OKR is a closed loop system residual slip is indeed very slow.

A reference to Liu, Huberman and Scanziani 2016 has been added in the Discussion (line 1749 "for speed selectivity in ON DSGCs"). Of course, we agree that there are visual neurons in many visual centers and in the retina itself that respond well to the fast speeds that ON DSGCs fail to respond to. We prominently feature ON-OFF DSGCs as an example of such a cell type and suggest that robust responses to fast motion derive in part from glutamatergic inputs from the same amacrine type as inhibits the ON DSGCs. These signals are carried forward to the superior colliculus and visual cortex. The same centers derive input from many other types of RGCs with strong responses to fast motion (see Fig. 1D). More specifically, the reviewer may be referring to the fact that some fast motion signals in the cortex are carried into the image stabilization system in monkeys and cats, which have frontalized eyes and large oculomotor ranges. These influences seem to be largely absent in rabbits and mice. The accessory optic system nuclei in mouse receive corticofugal projections from the visual cortex that have a role in plastic adaptation of the OKR gain (Scanziani 2016). These increase OKR gain at a bit higher speeds, but the temporal frequency response profile presented in Scanziani 2016 still fits well with the

response profile of ON DSGCs (the highest speeds in that work being 31 degrees/s or 1000 $\mu\text{m/s}$).

We agree with the reviewer that OKR can be evoked in the absence of head movements and the VOR (e.g., in head fixed mice), that OKR is a closed loop system, and that residual slip is very slow during head rotation. Length limits prevent us from squeezing these points into the text, which was already far too long.

Unfortunately is the outcome of the application of DREADD system a little unconvincing and it does not really add important information to the results with the strychnine application. In my judgement the DREADD section could be omitted. The authors have a difficult time to explain the not so perfect results and leave the reader in a puzzle.

We take the reviewer's point, but believe half a loaf is better than none. The DREADD effect, though incomplete, was significant and in the direction expected from our hypothesized circuit. This observation provides convergent evidence on the central point of the study, so we argue for keeping it in the paper. Readers will surely expect some explanation for the incompleteness of the DREADD effect but we agree that working through the possible scenarios disrupts the flow of the presentation. For this reason, we have moved the bulk of this discourse to Supplementary Information. The main text now merely flags the issue and directs the reader to the relevant Supplementary text.

It would hardly be surprising if the DREADD manipulation incompletely silenced VGLUT3 cells; such partial suppression has been observed in other neural systems. The messiness comes from the fact that we are obliged to consider the alternative possibility that the effect was incomplete because of the contribution of some other glycinergic input. We believe it is important for the reader to understand that we cannot definitively rule out such a contribution.

Also the electron microscopic reconstruction section interrupts the elegant flow of the physiological experiments and interpretation. May be it could be taken out here and serve as a building block for another more extensive morphological study and argumentation. I am absolutely not an expert on this topic and can give only a biased view.

We agree with the reviewer that the extensive passages on the SEM findings took us into secondary topics that disrupted the flow of the main story. However, we know that what we offered there was a substantial advance in our understanding of VGLUT3 mosaics and synaptic connectivity and will be of great interest to the retinal circuitry community. Our solution has been to trim this section in the main body of the text so as to present only the most cogent observations, while moving the bulk of this text and one full figure to a new

Supplementary section. We think this strikes the right balance between improving readability for the broad audience while still offering the aficionados access to the detailed anatomical findings.

These 2 omissions would focus the reader on the impressive physiological data and streamline the interpretation. I do not make this a condition sine qua non to highly recommend the ms for publication after minor review.

Much appreciated!

Specific comments:

Line 25: different motion speeds (instead of only fast)

Part of the significance of the work is that it has identified a role of VGluT3 dendrites specifically in modulating responses to fast motion in the retina, possibly in many retinal ganglion cell types. Therefore, we would like to keep the current phrasing.

59 there is a more direct link from the AOS to the oculomotor system through the nucleus prepositus hypoglossi

Fair enough. Indeed, the image-stabilization network other important nodes in the brainstem, not merely the prepositus. For clarity and to encompass this complexity, we modified the sentence in question to read: *“They send their axons almost exclusively to the nuclei of the accessory optic system, which relays their retinal slip signals to the vestibulocerebellum and brainstem oculomotor centers.”*

266 ff plexus plural is plexus not plexuses

Thank you, fixed.

218-328 aesthetically pleasing but not really important in the context the physiological study.

Thank you! Most of it has been transferred to Supplementary Information.

435 VGluT3 cells excites ??

Thank you, fixed.

Discussion: cortical input shown by Scanziani . Monkeys and cats

Added in line 1749 (“for speed selectivity in ON DSGCs”).

647 an interneuron

Thank you, fixed.

406 + 579+ 654 +664 Chemogenetic suppression is incomplete or only partly effective, could there be a speed tuned excitatory input ??

The partial effect on inhibition has been seen in voltage clamp experiments (Fig. 4A-C), where the holding voltage at the reversal potential of

excitation nulls excitation and isolates inhibition. We have added the words 'voltage clamp' in the legend of Fig. 4 and the holding voltages (and more data on Supplementary Fig. 7A, B). The question of whether there is VGlut3 contribution to the speed profile of excitation of ON DSGCs is in place, and the answer is that we haven't seen such an effect. We have added a line stating that (line 1329, "suppression of VGlut3 did not...").

606 slow retinal speed is also and probably mainly due to closed loop mechanisms. OKR reduces speed even further

We have removed the entire paragraph due to the considerations mentioned above (removal of unnecessary stress on OKN and text length).

640 saccades are not influenced by OKR but the reflex may stabilize the eye after saccades (Fred Miles)

We have removed the entire paragraph due to the considerations mentioned above.

711 that that

Thank you, fixed.

779 4 cardinal directions does not apply to ON DS cells What are the 4 on DS subtypes ???

We have changed the sentence to exclude mention of the number of preferred directions, now in line 2405 ("four cardinal directions"). It is true that it has been widely assumed that there were only three directions/subtypes until 2017. In a study published in 2017, we have carefully mapped retina-wide preferred directions in both ON, ON-OFF DSGCs and found that both classes had subtypes preferring the four cardinal directions, associated with optic flow during forward, backward, up and down motion of the animal (Sabbah et. al. *Nature* **546**, 492–497 (2017)).

787 is there any evidence

We have changed the phrasing (now lines 2427-2435, "If VGlut3 dendrites are orientation/direction selective, can they..."), making a weaker statement. The only evidence in favor of VGlut3 being capable of suppressing any preferred direction is that the inhibitory mechanism suppressing ON DSGCs seems similar in all ON-DSGCs we tested. We conducted experiments trying to find directionality in ON-DSGC inhibition in response to fast speed (Supplementary Fig. 11), but we did not observe strong directional preference.

REVIEWER COMMENTS

Reviewer #1 (Remarks to the Author):

The authors have adequately addressed my concerns. This is an excellent study.

Reviewer #2 (Remarks to the Author):

The authors have clarified concerns raised in the first round of review, with the addition of new data, statistical analysis, and updates to the figures.

One very puzzling inconsistency remains unresolved and that concerns the marked differences between the properties of the VGLuT3 AC Ca-responses (Fig. 5) and the glycinergic inhibition in On-DSGCs (Fig. 1). The authors now note that the orientation and direction-selectivity of VGLuT3 AC Ca-responses were not evident in the glycinergic inhibitory inputs to ON-DSGCs. Moreover, the response-delays (Fig. 5B) appear to be considerably longer than previously reported (ref 29). Resolving these issues would require a large amount of additional experimentation that would be beyond the scope of the present study.

Overall, the balance of evidence seems to support the fundamental finding that glycinergic inhibition from VGLuT3 ACs modulates speed tuning of ON-DSGCs. This is an interesting and novel finding that is likely to have significant impact in the field. More generally, the differing roles of excitation and inhibition, arising from the same cell, would be of broad interest to neuroscientists.

I have several additional points for clarification arising from the new data and analysis:

Line 139: is there a reference support the statement that an E/I ratio of 3 is sufficient?

Figs. 1, 4 & 5: A 1-tailed T-test was performed to test for significance. Were differences significant at all speeds? How were any effects of grating speed accounted for in the statistical analysis in the data?

Fig. 3H,I: Please provide more detail regarding the dotted line. Is it the mean variance of the baseline membrane current? What is the shading in H?

Line 290: Why the requirement for 1 sec LED flashes to optogenetically evoke EPSCs when the duration of the EPSCs appears to be less than 500 ms?

Line 293: in all 8 ON-DSGCs in which VGLuT3-activation evoked EPSCs, IPSCs were also observed. Given the discrepancy between these results and previous accounts that show only excitatory connections, it seems important to compare the sizes of the EPSCs and IPSCs in the 8 cells.

Lines 338 – 340: A new finding is presented, but no supporting data shown.

Lines 352 - 355: The significance of the speed dependence was raised in the first review. The revision now includes new analysis. Further details of the methods would be helpful. In determining the half-maximal speed, was the "descending branch" (L718) fit over a range of data-points or was it the interpolation between two data-points? Does an analysis of the variance, taking speed into account, support the contention that suppression is significantly stronger at higher speeds? The effects look modest, and the sample size is small.

Minor points:

Figs. 1,4 – None of the normalized curves appear to have a maximum of 1.

Line 145: Fig. 1B, red. Incorrect figure reference.

Line 157: remove comma, "... inhibition and dramatically...".

Line 733: Incorrect figure reference.

Fig. 3H: excitatory current is negative.

Fig. 5B: Use of color in B, C & D is potentially confusing as I was expecting some direct connection between the traces in B and the data in C & D. Suggest removing color from B.

Line 423: "suggest show"

Reviewer #3 (Remarks to the Author):

my comment:

All experiments are done in-vitro on the explanted retina. It is thus a little astonishing how much emphasis the authors put on the well established (not from the authors of this study) classic role of On Ds-cells for image-stabilizing eye-movements. It would be enough to mention that once in the introduction.

After reading the title and the first sentences in the introduction I was expecting experiment to show the behavioral effects of manipulating the retinal circuits on eye movements. But I was disappointed in this respect.

Author's response:

We are certainly sorry to disappoint! It was not for lack of trying. We had always hoped to include a behavioral result with these findings and established a collaboration with another group (J.C. Cang, Univ. of Virginia) to make the necessary measurements. Unfortunately, we had to rely on the DREADD approach to suppress the VGlut3 contribution. Our physiological data strongly suggest that the suppression of VGlut3 activity we achieved was relatively modest, and any effect on OKN may have been obscured by the inherent variability in the behavioral data. Since we couldn't deliver the behavioral result, we have revised the text to reduce the emphasis on this behavioral dimension of our work. We still frame our results in the context of the well-established role for the fast-speed inhibition of ON DSGCs in the optokinetic behavior of mammals. However, we have changed the title in an effort to signal that our study is focused on the retinal level. In addition, we have removed information/comments regarding OKN that were not directly relevant to the results of the current study:

line 12, 20 (abstract), lines 99 ("Animals see better..."), 1748 ("Optokinetic image stabilization is..."), 1867 ("Image stabilization is ubiquitous..."), 2047 ("The two classes...") in the introduction and discussion.

This is an appropriate correction

My comment:

... this is an impressive study on the retinal circuitry vetoing the response of ON DS-cells to higher velocities. However the visual system of many mammals goes to great effort to overcome this shortcoming of retinal ON DS-cells by having a visual cortex to provide responses to higher speeds. This may even be so in the mouse (but see Scanziani Nature 2016). Also it is not only the slip during VOR which is minimized by the OKR. Because the OKR is a closed loop system residual slip is indeed very slow.

Author's response

A reference to Liu, Huberman and Scanziani 2016 has been added in the Discussion (line 1749 "for speed selectivity in ON DSGCs"). Of course, we agree that there are visual neurons in many visual centers and in the retina itself that respond well to the fast speeds that ON DSGCs fail to respond to. We prominently feature ON-OFF DSGCs as an example of such a cell type and suggest that robust responses to fast motion derive in part from glutamatergic inputs from the same amacrine type as inhibits the ON DSGCs. These signals are carried forward to the superior colliculus and visual cortex. The same centers derive input from many other types of RGCs with strong responses to fast motion (see Fig. 1D). More specifically, the reviewer may be referring to the fact that some fast motion signals in the cortex are carried into the image stabilization system in monkeys and cats, which have frontalized eyes and large oculomotor ranges. These influences seem to be largely absent in rabbits and mice. The accessory optic system nuclei in mouse receive corticofugal projections from the visual cortex that have a role in plastic adaptation of the OKR gain (Scanziani 2016). These increase OKR gain at a bit higher speeds, but the temporal frequency response profile presented in Scanziani 2016 still fits well with the response profile of ON DSGCs (the highest speeds in that work being 31 degrees/s or 1000 $\mu\text{m/s}$). We agree with the reviewer that OKR can be evoked in the absence of head movements and the VOR (e.g., in head fixed mice), that OKR is a closed loop system, and that residual slip is very slow during head rotation. Length limits prevent us from squeezing these points into the text, which was already far too long.

Rebuttal accepted

My comment

Unfortunately is the outcome of the application of DREADD system a little unconvincing and it does not really add important information to the results with the strychnine application. In my judgement the DREADD section could be omitted. The authors have a difficult time to explain the not so perfect results and leave the reader in a puzzle.

Author's response

We take the reviewer's point, but believe half a loaf is better than none. The DREADD effect, though incomplete, was significant and in the direction expected from our hypothesized circuit. This observation provides convergent evidence on the central point of the study, so we argue for keeping it in the paper. Readers will surely expect some explanation for the incompleteness of the DREADD effect but we agree that working through the possible scenarios disrupts the flow of the presentation. For this reason, we have moved the bulk of this discourse to Supplementary Information. The main text now merely flags the issue and directs the reader to the relevant Supplementary text. It would hardly be surprising if the DREADD manipulation incompletely silenced VGLUT3 cells; such partial suppression has been observed in other neural systems. The messiness comes from the fact that we are obliged to consider the alternative possibility that the effect was incomplete because of the contribution of some other glycinergic input. We believe it is important for the reader to understand that we cannot definitively rule out such a contribution.

Acceptable after moving the bulk of this discourse to Supplementary Information.

My comment

Also the electron microscopic reconstruction section interrupts the elegant flow of the physiological experiments and interpretation. May be it could be taken out here and serve as a building block for another more extensive morphological study and argumentation. I am absolutely not an expert on this topic and can give only a biased view.

Author's response

We agree with the reviewer that the extensive passages on the SEM findings took us into secondary topics that disrupted the flow of the main story. However, we know that what we offered there was a substantial advance in our understanding of VGLUT3 mosaics and synaptic

connectivity and will be of great interest to the retinal circuitry community. Our solution has been to trim this

section in the main body of the text so as to present only the most cogent observations, while moving the bulk of this text and one full figure to a new Supplementary section. We think this strikes the right balance between improving readability for the broad audience while still offering the aficionados access to the detailed anatomical findings.

Good move

These 2 omissions would focus the reader on the impressive physiological data and streamline the interpretation. I do not make this a condition sine qua non to highly recommend the ms for publication after minor review.

Author's response:

Much appreciated!

You're welcome

Specific comments:

Line 25: different motion speeds (instead of only fast)

Part of the significance of the work is that it has identified a role of VGlut3 dendrites specifically in modulating responses to fast motion in the retina, possibly in many retinal ganglion cell types. Therefore, we would like to keep the current phrasing.

Acceptable

59 there is a more direct link from the AOS to the oculomotor system through the nucleus prepositus hypoglossi

Fair enough. Indeed, the image-stabilization network other important nodes in the brainstem, not merely the prepositus. For clarity and to encompass this complexity, we modified the sentence in question to read: "They send their axons almost exclusively to the nuclei of the accessory optic system, which relays their retinal slip signals to the vestibulocerebellum and brainstem oculomotor centers."

As requested

266 ff plexus plural is plexus not plexuses

Thank you, fixed.

218-328 aesthetically pleasing but not really important in the context of the physiological study.

Thank you! Most of it has been transferred to Supplementary Information.

o.k.

435 VGlut3 cells excites ??

Thank you, fixed.

Discussion: cortical input shown by Scanziani . Monkeys and cats

Added in line 1749 ("for speed selectivity in ON DSGCs").

Ok

647 an interneuron

Thank you, fixed.

406 + 579+ 654 +664 Chemogenetic suppression is incomplete or only partly effective, could there be a speed tuned excitatory input ??

Author's response

The partial effect on inhibition has been seen in voltage clamp experiments (Fig. 4A-C), where the holding voltage at the reversal potential of excitation nulls excitation and isolates inhibition. We have added the words 'voltage clamp' in the legend of Fig. 4 and the holding voltages (and more data on Supplementary Fig. 7A, B). The question of whether there is VGlut3 contribution to the speed profile of excitation of ON DSGCs is in place, and the answer is that we haven't seen such an effect. We have added a line stating that

ok

(line 1329, "suppression of VGlut3 did not...").

606 slow retinal speed is also and probably mainly due to closed loop mechanisms. OKR reduces speed even further

We have removed the entire paragraph due to the considerations mentioned above (removal of unnecessary stress on OKN and text length).

Fine

640 saccades are not influenced by OKR but the reflex may stabilize the eye after saccades (Fred Miles)

We have removed the entire paragraph due to the considerations mentioned above.

ok

711 that that

Thank you, fixed.

779 4 cardinal directions does not apply to ON DS cells What are the 4 on DS subtypes ???

We have changed the sentence to exclude mention of the number of preferred directions, now in line 2405 ("four cardinal directions"). It is true that it has been widely assumed that there were only three directions/subtypes until 2017. In a study published in 2017, we have carefully mapped retina-wide preferred directions in both ON, ON-OFF DSGCs and found that both classes had subtypes preferring the four cardinal directions, associated with optic flow during forward, backward, up and down motion of the animal (Sabbah et. al. Nature 546, 492-497 (2017)).

That's correct. I should have remembered your important study

787 is there any evidence

We have changed the phrasing (now lines 2427-2435, "If VGlut3 dendrites are orientation/direction selective, can they..."), making a weaker statement. The only evidence in favor of VGlut3 being capable of suppressing any preferred direction is that the inhibitory mechanism suppressing ON DSGCs seems similar in all ON-DSGCs we tested. We conducted experiments trying to find directionality in ON-DSGC inhibition in response to fast speed (Supplementary Fig. 11), but we did not observe strong directional preference

accepted

REVIEWER COMMENTS

Reviewer #2 (Remarks to the Author):

The authors have clarified concerns raised in the first round of review, with the addition of new data, statistical analysis, and updates to the figures.

One very puzzling inconsistency remains unresolved and that concerns the marked differences between the properties of the VGlut3 AC Ca-responses (Fig. 5) and the glycinergic inhibition in On-DSGCs (Fig. 1). The authors now note that the orientation and direction-selectivity of VGlut3 AC Ca-responses were not evident in the glycinergic inhibitory inputs to ON-DSGCs. Moreover, the response-delays (Fig. 5B) appear to be considerably longer than previously reported (ref 29). Resolving these issues would require a large amount of additional experimentation that would be beyond the scope of the present study.

We have edited the relevant discussion paragraph to flag this discrepancy for readers (line 610):

“This delayed VGlut3 Ca²⁺ response contrasts with the brisk inhibitory currents evoked by the same stimuli in postsynaptic ON-DSGCs (Fig. 1C). In this context, it must be remembered that calcium signals could misrepresent true membrane voltage due to slow kinetics, nonlinearities, and insensitivity to near-threshold voltage changes. However, a fast contribution to inhibitory currents in ON-DSGCs from ON SACs cannot be ruled out.”

Our findings are in apparent conflict with the strong surround suppression seen in vGlut3 cells by several groups and their role in object motion sensing as proposed by Kerschensteiner’s group (now refs. 33, 34). This discrepancy prompted us to conduct new experiments with another method (calcium imaging). These experiments yielded the surprising and remarkable findings that surround suppression in vGlut3 is stimulus-dependent, and that vGlut3 cells consistently respond to global motion. Thus, our study demonstrates that these amacrine cells have appropriate visual responses for the role we ascribe to them, though further exploration of the apparent discrepancy would certainly be welcome.

As requested by Reviewer #2 in the first round of review, we assessed orientation tuning in the postsynaptic currents. The inconsistencies mentioned between the presynaptic calcium responses and the postsynaptic currents indeed remain a mystery, and could arise in many ways. As the reviewer states, this would take much more work to resolve and is beyond the scope of this paper. We have acknowledged and explained this issue to the best of our ability following the previous and current reviews, in the two last paragraphs of the discussion (Line 601-625).

Overall, the balance of evidence seems to support the fundamental finding that glycinergic inhibition from VGLUT3 ACs modulates speed tuning of ON-DSGCs. This is an interesting and novel finding that is likely to have significant impact in the field. More generally, the differing roles of excitation and inhibition, arising from the same cell, would be of broad interest to neuroscientists.

I have several additional points for clarification arising from the new data and analysis:

Line 139: is there a reference support the statement that an E/I ratio of 3 is sufficient?

The statement has been modified to refer to ON-OFF DSGCs as a close example, and references were added (Line 140):

“In ON-OFF DSGCs, inhibition $\sim 3 - 3.5$ times as large as excitation is sufficient to suppress RGC firing^{23,24}. “

Figs. 1, 4 & 5: A 1-tailed T-test was performed to test for significance. Were differences significant at all speeds? How were any effects of grating speed accounted for in the statistical analysis in the data?

We have now tested and added the results of the t-test at every speed in Figs. 1F, 1G, 4A-C, 4F, 4G, 5F, 5G. We have also kept the previous tests that refer to effects detailed in the text. Details have been added in the figure legends and in the Methods section regarding this.

Fig. 3H,I: Please provide more detail regarding the dotted line. Is it the mean variance of the baseline membrane current? What is the shading in H?

The dotted line is the average of the noise level over cells, where the “noise level” has been taken as $2 \times$ (the standard deviation of the baseline current). These details have been explained in the methods section (“The noise level for each cell...”, Lines 745-750). This passage has been modified in an effort to make it clearer. Another reference to “Methods” has been added to the legend of Fig. 3.

“The noise level for each cell (and each holding voltage) was taken as the standard deviation of the smoothed current trace during the pre-stimulus interval, multiplied by 2. In Fig. 3B, C, E, H and I, this noise level was averaged over cells (denoted by a dotted line). The shaded regions in Fig. 3B, C, E, H denote the resulting average \pm SD. In Fig. 3C, E, and I, currents and noise levels were normalized by the maximal current in the control condition.”

Line 290: Why the requirement for 1 sec LED flashes to optogenetically evoke EPSCs when the duration of the EPSCs appears to be less than 500 ms?

We have used two widely used optogenetic stimulus durations for these experiments, 100ms and 1s. Excitatory currents were very modest and often ramped gradually towards the maximal inward current; the response shown in Fig. 3G is atypical in this regard. We have added more typical current traces in Supplementary Fig. 5E in order to illustrate this. We suspect that this is why we have not detected any excitation in the 100ms trials.

Line 293: in all 8 ON-DSGCs in which VGluT3-activation evoked EPSCs, IPSCs were also observed. Given the discrepancy between these results and previous accounts that show only excitatory connections, it seems important to compare the sizes of the EPSCs and IPSCs in the 8 cells.

We have added this data in Supplementary Fig. 5D (with example traces of EPSCs and IPSCs in the same cells in Supplementary Fig. 5E). A reference to this data was made in the main text, line 302:

“Optogenetically evoked IPSC and EPSC measured in the same cells correlated positively (Supplementary Fig. 5D, E; $R^2 = 0.53$, $n = 8$ cells). This is consistent with both currents arising from VGluT3 cells and variable success at driving that dual input optogenetically.”

Lines 338 – 340: A new finding is presented, but no supporting data shown.

We have added the relevant data in Supplementary Fig.8C, D. The figure legend has been updated.

Lines 352 - 355: The significance of the speed dependence was raised in the first review. The revision now includes new analysis. Further details of the methods would be helpful. In determining the half-maximal speed, was the “descending branch” (L718) fit over a range of data-points or was it the interpolation between two data-points?...

The Methods section has been updated following the first review and again this time to include new analysis (see specific points in this review and the previous one). The “descending branch” was calculated with the curves linearly interpolated between data points (Methods: line 732):

“The ‘half maximum speed’ was the speed at which the descending branch of the curve, as determined by linear interpolation between data points, crossed the horizontal line of half the maximal response, and was considered a cutoff speed for the cell’s responses.”

...Does an analysis of the variance, taking speed into account, support the contention that suppression is significantly stronger at higher speeds? The effects look modest, and the sample size is small.

Applying the t-test to data points at different speeds, statistical significance was achieved only at the data points shown in the figure panels 4F, G. We therefore accept the reviewer's perspective on this point, and have removed the statement that the reduction was more pronounced at faster speeds than at slower speeds, as well as the comparison between curves along the line of half maximal response.

Minor points:

Figs. 1,4 – None of the normalized curves appear to have a maximum of 1.

The curve for each individual cell in the population has been normalized so that its maximum response was 1. Since these often occurred at different speeds (x-values) for different cells, the sum curve is <1 for any x-value. The process is explained in Methods, Lines 727, 735.

“In population response vs. speed data, curves from different cells were normalized by their maximum and averaged. If different cells in the population had data points at different speeds, their interpolated curves were averaged...

In population data where synaptic blockers or CNO were used (DREADDs), the response curves for each cell were normalized by the maximum of the control curve for that cell, and then curves were averaged over cells.”

Line 145: Fig. 1B, red. Incorrect figure reference.

Thank you, fixed.

Line 157: remove comma, “... inhibition and dramatically...”.

Thank you, fixed.

Line 733: Incorrect figure reference.

Thank you, fixed.

Fig. 3H: excitatory current is negative.

“Inward current” (Fig. 3H, I) and “outward current” (Fig. 3B, C, E) have been added to the axis titles and figure legend for accuracy in terminology.

Fig. 5B: Use of color in B, C & D is potentially confusing as I was expecting some direct connection between the traces in B and the data in C & D. Suggest removing color from B.

The color code used here (Fig. 5B) is similar to the one used elsewhere in the manuscript (e.g. Fig. 1, 4), with the same color showing both the population data (ROIs in this case), and a trace from a single example. We would like to keep this color code here to retain the information. However, to prevent the potential confusion between ROIs and FOVs, In Fig. 5C we have also colored the specific ROI curve corresponding to the data in B. (Average over ROIs: thick blue line, single ROI: thin blue line). We have repeated this strategy in 5I. The figure legend has been updated to explain this.

Line 423: “suggest show”

Thank you, fixed.